# CELF1 is a central node in post-transcriptional regulatory programmes underlying EMT

Arindam Chaudhury[1,2], Shebna Cheema[1], Joseph M. Fachini[1,2], Natee Kongchan[1,2], Guojun Lu[1,2], Lukas M. Simon[3,4], Tao Wang[5], Sufeng Mao[5], Daniel G. Rosen[6], Michael M. Ittmann[6], Susan G. Hilsenbeck[2,5], Chad A. Shaw[3] & Joel R. Neilson[1,2]

The importance of translational regulation in tumour biology is increasingly appreciated. Here, we leverage polyribosomal profiling to prospectively define translational regulatory programs underlying epithelial-to-mesenchymal transition (EMT) in breast epithelial cells. We identify a group of ten translationally regulated drivers of EMT sharing a common GU-rich cis-element within the 3′-untranslated region (3′-UTR) of their mRNA. These cis-elements, necessary for the regulatory activity imparted by these 3′-UTRs, are directly bound by the CELF1 protein, which itself is regulated post-translationally during the EMT program. CELF1 is necessary and sufficient for both mesenchymal transition and metastatic colonization, and CELF1 protein, but not mRNA, is significantly overexpressed in human breast cancer tissues. Our data present an 11-component genetic pathway, invisible to transcriptional profiling approaches, in which the CELF1 protein functions as a central node controlling translational activation of genes driving EMT and ultimately tumour progression.

[1] Department of Molecular Physiology and Biophysics, Baylor College of Medicine, Houston, Texas 77030, USA. [2] Dan L. Duncan Cancer Center, Baylor College of Medicine, Houston, Texas 77030, USA. [3] Department of Molecular and Human Genetics, Baylor College of Medicine, Houston, Texas 77030, USA. [4] Department of Structural and Computational Biology and Molecular Biophysics, Baylor College of Medicine, Houston, Texas 77030, USA. [5] Lester and Sue Smith Breast Center, Baylor College of Medicine, Houston, Texas 77030, USA. [6] Department of Pathology, Baylor College of Medicine, Houston, Texas 77030, USA. Correspondence and requests for materials should be addressed to J.R.N. (email: neilson@bcm.edu).

Tumour metastasis underlies over 90% of cancer mortality[1]. In the invasion and metastasis cascade, cancer cells disseminate from a primary tumour to anatomically distant sites, eventually forming macrometastatic tumours[2]. The transdifferentiation of epithelial cells into motile mesenchymal cells, termed epithelial–mesenchymal transition (EMT), is central to the pathophysiology of tumour metastasis and cancer progression[3]. A myriad of studies have described the signalling pathways and associated transcriptional responses underlying EMT[2,3]. In comparison, the post-transcriptional responses contributing to the EMT program are less well understood. Consistent with reports demonstrating the widespread role of post-transcriptional regulation in gene expression and function[4], two themes have emerged regarding the role of translational control in other aspects of carcinogenesis[5,6]. First, under conditions of stress, cancer cells limit translation to a subset of proteins that promote cell survival. Second, increased levels of the proteins required to initiate translation releases a level of control on important modulators of the cell cycle, which leads to uncontrolled growth. Thus, global programs of translational control contribute both to the survival and the proliferation of cancer cells. It is thus reasonable to posit that translational programs similarly impact EMT and cancer metastasis. Consistent with this notion, recent findings have demonstrated that coordinated changes in post-transcriptional regulatory networks profoundly alter cellular phenotype and behaviour[7–9]. The epithelial phenotype is also regulated by microRNAs, most notably the miR-200 family and miR-205 (ref. 10).

To prospectively and functionally identify additional translational regulatory programs underlying EMT, we leveraged polyribosome enrichment/depletion analysis via next-generation sequencing to define translational control programs during EMT in a breast epithelial cell model. Our results define and genetically order an 11-member post-transcriptional regulatory circuit underlying breast cancer progression in which CELF1 (CUG RNA-binding protein and embryonically lethal abnormal vision-type RNA-binding protein 3-like factor 1) functions as a central regulator.

## Results

**Identification of translationally regulated genes in EMT.** To define translational programs governing EMT, we sought to identify mRNAs that are polysomally enriched or depleted in the epithelial and mesenchymal states. The MCF7 and MCF10A breast epithelial cell lines exhibit characteristics of normal mammary epithelial cells in monolayer cultures, and robust expression of E-cadherin (Fig. 1a,b). On treatment with transforming growth factor-β (TGF-β), MCF10A cells undergo EMT, characterized by loss of cell–cell contacts, the emergence of spindle-shaped fibroblast-like mesenchymal cells and induction of expression of mesenchymal cell markers, such as N-cadherin, fibronectin and vimentin. However, although the TGF-β signalling pathway is both intact and functional in MCF7 cells[11], these cells do not undergo EMT when treated with TGF-β (Fig. 1a,b). We rationalized that any event commonly observed in both cell lines could not be associated with the differential EMT response in these models (Supplementary Fig. 1a).

Post-nuclear extracts from biological triplicates of untreated and TGF-β-treated MCF7 and MCF10A cells were subjected to polyribosomal fractionation. Puromycin release[12] (Fig. 1c), analysis of ribosomal RNA occupancy[13] (Supplementary Fig. 1b), and immunoblot detection of eIF3C (eukaryotic initiation factor 3C) and rPS6 (ribosomal protein S6) in the lighter, non-polysomal fractions[14] (Fig. 1d) confirmed the fidelity of our fractionation. Poly(A) RNA isolated from both from pooled polysomal fractions and unfractionated post-nuclear extracts (total mRNA) were used to generate cDNA libraries for next-generation sequencing. We calculated enrichment or depletion of polyribosome-associated mRNA in each fraction relative to total cellular mRNA (Supplementary Data 1,2), and plotted these data in terms of mesenchymal against epithelial polyribosomal enrichment/depletion in both cell lines (Fig. 1e, Supplementary Data 3). Messenger RNA species subject to differential translational regulation in this context were defined as those (i) exhibiting polyribosomal enrichment or depletion with a post-corrected Storey q-value of < 0.03 in at least one of our four conditions in the initial analysis, and (ii) having a fold change of a minimum of two s.d. removed from the median difference in polyribosomal enrichment or depletion of all mRNAs in the two cell lines when the aggregate data set was compared. Via these criteria, we identified 200 mRNAs characterized by altered polyribosomal occupancy. Of these 200 mRNAs, 72 were putative positive regulators of EMT, while 128 were putative negative regulators. Among the 72 putative positive regulators was Snail (SNAI1), a transcription factor widely known to drive EMT[3]. We validated changes in the relative polysomal representation of randomly chosen mRNA transcripts in MCF10A cells via quantitative real-time PCR (qRT-PCR; Fig. 1f).

**A common 3′-UTR-motif in polysomally enriched mRNAs.** The potential contribution of a given transcript′s 3′-UTR to the stability and translation of the mRNA is well established. We thus used the MEME[15] and BioProspector[16] algorithms to identify enriched 6–10 nucleotide motifs within the 3′-UTRs of polysomally enriched and depleted transcripts. Both analyses revealed strong enrichment (MEME $P = 3.1e − 34$) for a distinct 5′-GTGTGTGTGT-3′ motif, or guanine/uridine-rich element (GRE) only in the positive regulator class. This motif was spread 43 times among 20 (27.8%) of the 72 putative positive regulators (Fig. 2a).

**GREs confer differential translational regulation.** We speculated that these GREs might function as a common cis-regulatory element. We first confirmed that changes in relative total mRNA and protein levels were uncoupled for these genes. Immunoblot analysis revealed striking increases in relative protein expression of 13 of 14 candidates tested on treatment of MCF10A (but not MCF7) cells, with TGF-β (Fig. 2b). Strikingly, analysis of total mRNA from the untreated and treated MCF10A cells revealed that except for two cases (SNAI1 and TICAM2), these increases in relative protein expression occurred independently of robust positive changes in mRNA expression (Fig. 2b).

We next utilized a bi-fluorescent reporter system[17] to ask whether the 3′-UTRs of the putative regulators were sufficient to confer differential expression in the mesenchymal state. We amplified 3′-UTRs encoded within 33 of the mRNAs present within our polysomally enriched set, including 14 GRE-containing 3′-UTRs (Fig. 1e), from MCF10A genomic DNA via PCR. We also amplified 3′-UTRs encoded within 37 mRNAs randomly chosen from our polysomally depleted set. This collection of UTRs, together with positive and negative control 3′-UTRs derived from the ZEB1 gene, were individually recombineered into our vector downstream of a turbo-RFP (tRFP) reporter coding sequence. The ZEB1 3′-UTR, which confers repression in the epithelial state, is progressively released from this repression as miR-200 levels decrease during EMT programs[10]. A mutant version of the ZEB1 3′-UTR gene, in which miR-200 family recognition sites have been ablated, is not subject to this control[10].

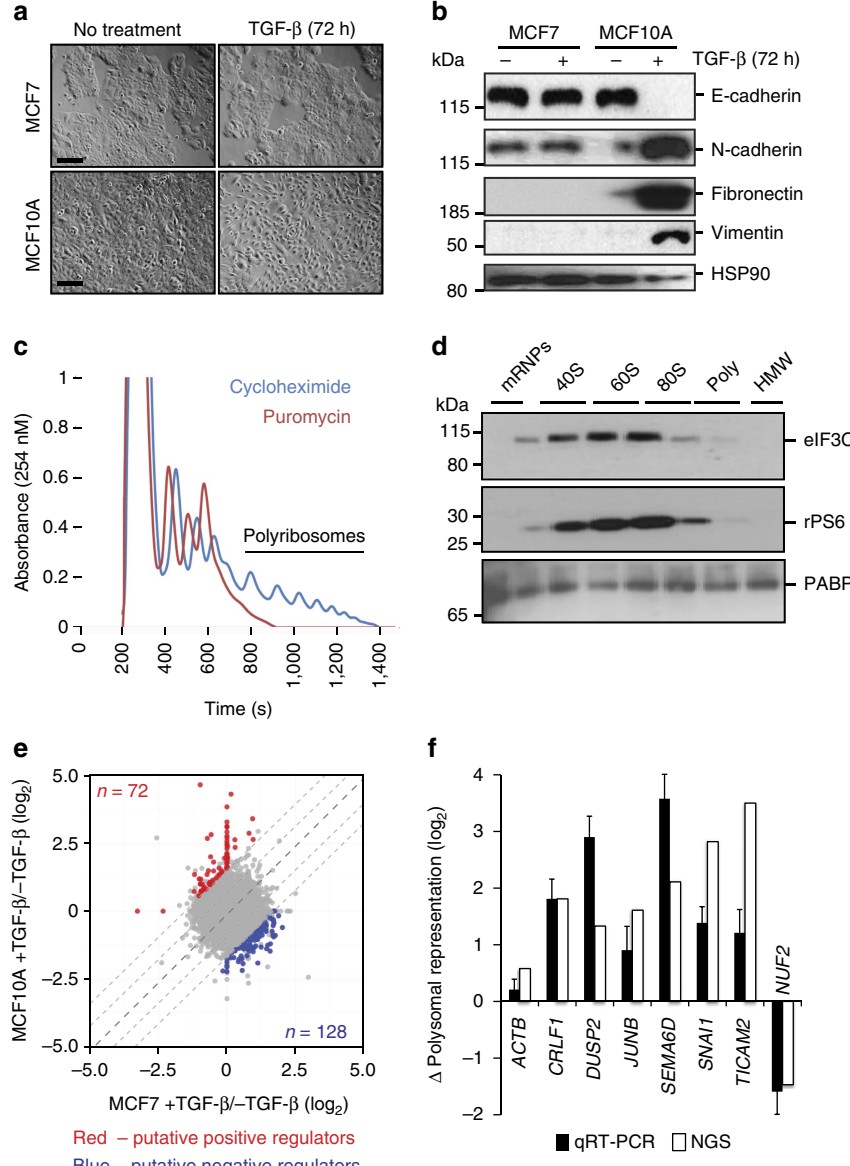

**Figure 1 | Polyribosomal profiling of MCF10A and MCF7 cells to identify translationally regulated genes in EMT.** (**a,b**) Phase-contrast micrographs (**a**) and immunoblot analysis of epithelial and mesenchymal markers (**b**) of untreated or TGF-β-treated MCF7 and MCF10A cells. Scale bar, 100 μm. Blots were stripped and re-probed for HSP90 (bottom panel) as a loading control. (**c,d**) Representative polyribosome isolation profile (**c**) and immunoblot (**d**) to demonstrate fidelity of fractionation. (**e**) Polyribosomal enrichment and depletion associated with EMT. On each axis, values derived for the indicated cell line treated with TGF-β are normalized to values derived from the same cell line in the absence of treatment. Center diagonal indicates mean of comparison, middle diagonals indicate one s.d. from the mean, outer diagonals indicate two s.d. from the mean. (**f**) qRT-PCR validation of polyribosomal enrichment and depletion of representative events from (**e**) using total and polyribosomal mRNA in untreated and TGF-β-treated MCF10A cells. All panels are representative of a minimum of three experimental replicates. For immunoblots depicted, samples were derived from the same experiment and gels were processed in parallel. Error bars depict s.e.m. See also Supplementary Fig. 1. Full scans of blots are shown in Supplementary Fig. 7. HMW, high molecular weight; NGS, next-generation sequencing.

tRFP and control turbo-GFP (tGFP) expression in TGF-β-treated and untreated samples were assessed via flow cytometry. EMT in the TGF-β-treated duplicates was verified both by visual examination and via monitoring of E-cadherin expression on the surface of each cell line during the flow cytometric analysis. We identified 14 GRE-containing 3′-UTR elements, conferring a more than or equal to twofold relative increase in normalized tRFP expression in mesenchymal MCF10A cells as compared with the epithelial state (Fig. 2c). The fifteenth GRE-containing UTR, derived from the *RHOB* gene, conferred no detectable change in tRFP expression (Fig. 2c).

We next asked whether the increased expression of these reporters in the mesenchymal state was conferred by the GREs within their associated 3′-UTRs. Indeed, deletion of the GRE markedly reduced or eliminated the increase in tRFP expression observed in TGF-β-treated cells (Fig. 2d). Differential mRNA expression or stability in the two states did not contribute to differences in reporter expression, as no significant changes in the relative mRNA expression of these reporters was observed (Fig. 2e). Cumulatively, these data indicate that the GRE contained within the 3′-UTR of a discrete cohort of mRNAs can confer increased relative protein expression in the context of TGF-β-induced EMT.

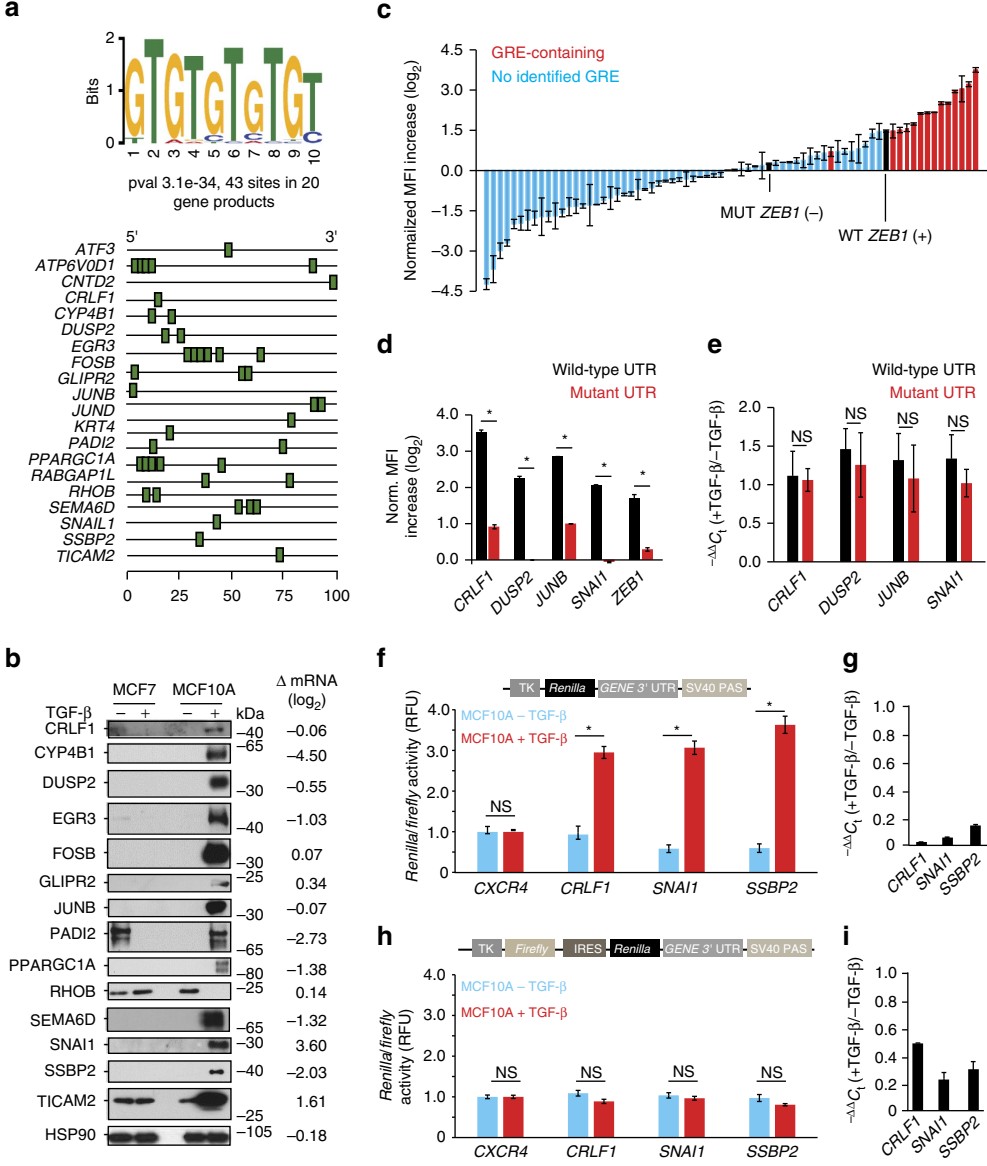

**Figure 2 | GREs confer context-dependent translational regulation to mRNA transcripts.** (**a**) MEME algorithm output of an enriched GRE within the 3′-UTRs of the putative positive regulator set and schematic representation of GRE location in 3′-UTR of indicated transcripts. (**b**) Immunoblot analysis of proteins encoded by GRE-containing mRNAs in untreated and TGF-β-treated MCF7 and MCF10A cells. HSP90 serves as a representative loading control. The relative changes in total mRNA expression (MCF10A ± TGF-β) of each GRE-containing gene in total mRNA as quantified by qRT-PCR are indicated on the right. (**c**) Reporter assay quantifying the relative tRFP expression from 70 3′-UTR reporters in TGF-β-treated MCF10A cells relative to untreated controls. Wild-type (WT) and mutant (MUT) *ZEB1* UTRs (black) serve as positive and negative controls, respectively. Data were normalized to relative tGFP expression. (**d**) Reporter assay quantifying the relative tRFP expression from indicated WT 3′-UTRs (black) and those in which the GRE (or miR-200 binding sites for *ZEB1*) (red) have been deleted in TGF-β-treated MCF10A cells relative to untreated controls. Data were normalized to relative tGFP expression. (**e**) qRT-PCR of relative mRNA expression of indicated WT and GRE-mutant tRFP reporters in untreated and TGF-β-treated MCF10A cells. (**f**) Reporter assay quantifying the relative *Renilla* luciferase expression from the indicated 3′-UTR luciferase reporters in TGF-β-treated MCF10A cells relative to untreated controls. *Renilla* activity was normalized to *Firefly* luciferase activity and this ratio is expressed relative to the parental control reporter (*pRL-TK CXCR4 6x*). (**g**) qRT-PCR of relative mRNA expression of indicated Renilla *Luciferase* reporters (pRL-TK) in untreated and TGF-β-treated MCF10A cells. (**h**) Reporter assay quantifying the relative *Renilla* luciferase expression from the indicated 3′-UTR luciferase reporters driven from an internal ribosomal entry site in TGF-β-treated MCF10A cells relative to untreated controls. *Renilla* activity was normalized to *Firefly* luciferase activity. Data are normalized to a control reporter containing the *pRL-TK CXCR4 6x* 3′-UTR. (**i**) qRT-PCR of relative mRNA expression of indicated *Renilla* luciferase reporters driven by EMCV IRES (pFR-EMCV) in untreated and TGF-β-treated MCF10A cells. All panels (excluding **a**) are representative of a minimum of three experimental replicates. For immunoblots depicted, samples were derived from the same experiment and gels were processed in parallel. Error bars in **d** depict s.d. of the mean, error bars in **e,f,h** depict s.e.m. NS indicates non-significant, *$P \leq 0.05$, (Student's *t*-test). Full scans of blots are shown in Supplementary Fig. 8.

To assess the potential contributions of translational initiation and translational elongation to the observed increases in protein expression, we fused the 3′-UTRs of a subset of GRE-containing mRNAs downstream of the *Renilla* luciferase-coding sequence in the pRL-TK-CXCR4-6x reporter plasmid[18] (Fig. 2f). We built a parallel battery of bicistronic constructs in which the same thymidine kinase promoter was utilized to drive expression of the firefly luciferase-coding sequence, followed by the EMCV

internal ribosome entry site (IRES), the *Renilla* luciferase open-reading frame and individual GRE-containing 3′-UTRs (pFR-EMCV; Fig. 2h).

Transfection of the GRE-containing 3′-UTR reporters into untreated and TGF-β-treated MCF10A cells revealed a significant increase in reporter activity specific to treated cells (Fig. 2f), again independent of any differences in relative mRNA expression (Fig. 2g). In contrast, comparison of IRES-driven *Renilla* luciferase expression in the bicistronic pFR-EMCV reporters revealed no differences in relative protein and mRNA expression between the epithelial and mesenchymal states (Fig. 2h,i). These results are consistent with a model in which translational control of mRNAs harbouring GREs within their 3′-UTRs is mediated at the level of 5′ $^{7m}$G cap-dependent translational initiation.

**GRE-containing mRNAs are necessary and sufficient for EMT.** To determine whether the GRE-containing mRNAs functionally contribute to EMT, we transiently transfected MCF10A cells with siRNAs targeting *Firefly* luciferase (negative control) or each of the 13 GRE-containing transcripts, and treated the transfectants with TGF-β for 72 h (Supplementary Fig. 2a). After 72 h, EMT of the transfectants was monitored via immunoblot analysis of molecular markers, and migration and invasion in standard transwell assays. Via these criteria, 10 of the 13 GRE-containing mRNAs tested blocked TGF-β-induced EMT (Fig. 3a,b, Supplementary Fig. 2b), indicating that these mRNAs are necessary for TGF-β-induced EMT. Importantly, these effects were independent of changes in cellular proliferation or apoptosis (Supplementary Fig. 2c,d).

We next asked whether any of the proteins encoded by the GRE-containing mRNAs were sufficient to drive EMT independent of TGF-β treatment. MCF10A cells were individually transduced with *pLenti6.3* vectors encoding *Renilla*

luciferase or each of the 13 individual coding sequences derived from the GRE-containing mRNAs (Supplementary Fig. 2a). Using the same criteria described above, our results revealed that 6 (*EGR3*, *FOSB*, *PADI2*, *SEMA6D*, *SNAI1* and *SSBP2*) of the 13 proteins encoded by GRE-containing mRNAs tested are sufficient to drive EMT independent of TGF-β (Fig. 3c,d, Supplementary Fig. 2e). Confounding effects on proliferation or viability of individual transduced lines were again ruled out (Supplementary Fig. 2f,g).

**CELF1 regulates GRE-containing EMT driver mRNAs.** A conserved *cis*-element within 3′-UTRs of a discrete set of EMT-inducer mRNAs suggests that a common *trans*-regulator integrates their control. A literature search revealed the GRE motif as a putative binding site for the CELF (CUGBP and embryonically lethal abnormal vision-type RNA-binding protein 3-like factor) or muscleblind-like splicing regulator (MBNL) families of RNA-binding proteins[19]. Immunoblot analysis of CELF1, CELF2, MBNL1 and MBNL2 in untreated, and TGF-β-treated MCF10A and MCF7 cells revealed specific and marked upregulation of CELF1, MBNL1 and MBNL2 in MCF10A cells in response to TGF-β (Fig. 4a).

To assess the putative functional requirement of CELF and MBNL proteins in EMT, we transiently transfected MCF10A cells with siRNAs targeting *Firefly* luciferase, *CELF1*, *CELF2*, *MBNL1* or *MBNL2*. Transfectants were treated with TGF-β, and assessed for EMT. Knockdown of *CELF1* alone blocked TGF-β-mediated EMT, and this did not materially impact cell proliferation or viability (Fig. 4b,c, Supplementary Fig. 3a–d), indicating that *CELF1* is necessary for TGF-β-induced EMT in MCF10A cells. We next asked whether *CELF1* misexpression was sufficient to induce EMT in untreated MCF10A cells. Cells transfected with the *CELF1* overexpression construct, but not mock transfectants,

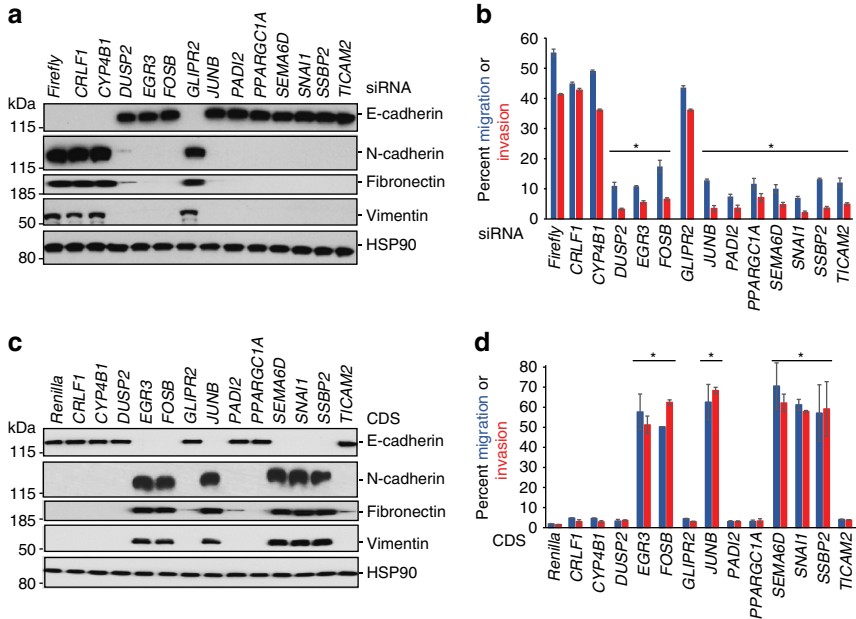

**Figure 3 | GREs encode gene products required for EMT.** (**a**) Immunoblot analysis of epithelial and mesenchymal cell markers in TGF-β-treated MCF10A cells transiently transfected with siRNAs targeting Firefly *Luciferase* or the indicated mRNAs. HSP90 serves as a loading control. (**b**) Quantification of relative cellular migration and invasion in transwell assays in TGF-β-treated MCF10A cells transiently transfected with siRNAs targeting Firefly *Luciferase* or the indicated mRNAs. (**c,d**) As in **a** and **b** except that untreated MCF10A cells were stably transduced with pL6.3-Renilla *Luciferase* or pL6.3-driving the expression of the indicated coding sequences, respectively. All panels are representative of a minimum of three experimental replicates. For immunoblots depicted, samples were derived from the same experiment and gels were processed in parallel. Error bars depict s.d. of the mean. *$P \leq 0.05$ (Student's *t*-test). See also Supplementary Fig. 2. Full scans of blots are shown in Supplementary Fig. 9.

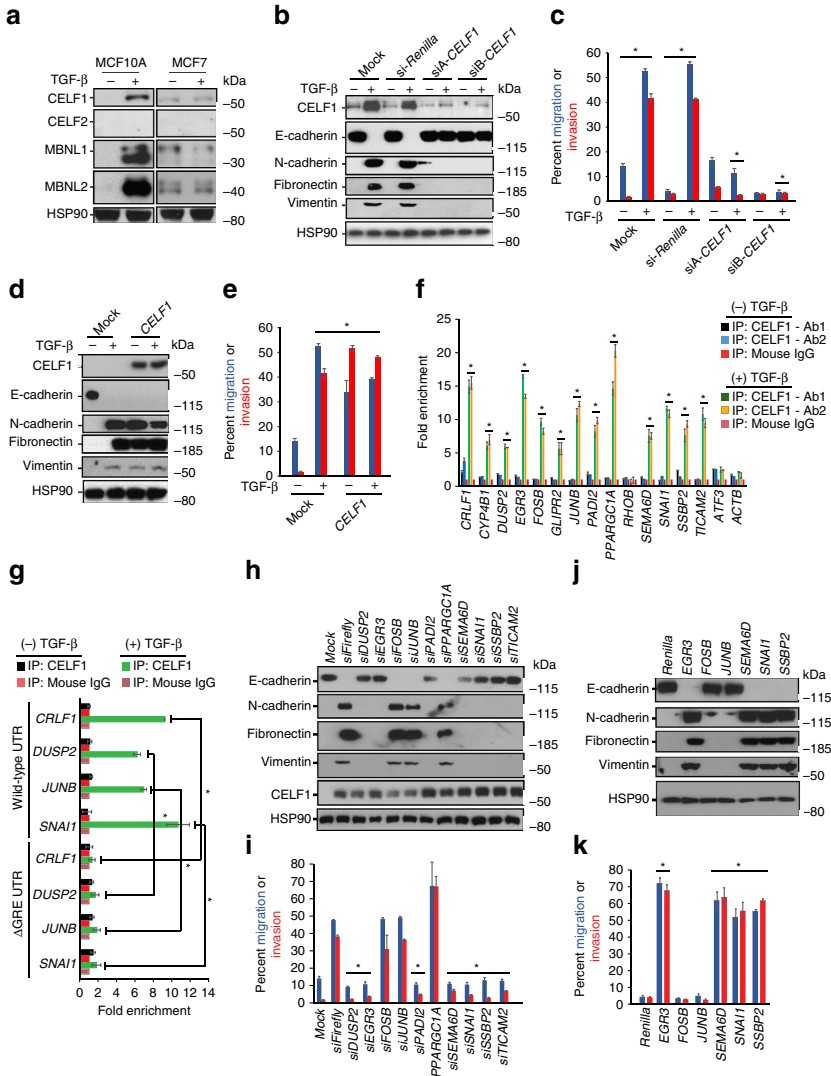

**Figure 4 | CELF1 is a major regulator of GRE-containing mRNAs encoding drivers of EMT.** (**a**) Immunoblot analysis of CELF and MBNL proteins in untreated and TGF-β-treated MCF7 and MCF10A cells. HSP90 serves as a loading control. (**b**) Immunoblot analysis of epithelial and mesenchymal markers in untreated and TGF-β-treated MCF10A cells transiently transfected with the indicated siRNAs. HSP90 serves as a loading control. (**c**) Quantification of relative cellular migration and invasion in transwell assays in untreated and TGF-β-treated MCF10A cells transiently transfected with the indicated siRNAs or overexpression constructs. (**d**) Immunoblot analysis of epithelial and mesenchymal markers in untreated and TGF-β-treated MCF10A cells transiently transfected with vehicle or *CELF1* overexpression construct. HSP90 serves as a loading control. (**e**) Quantification of relative cellular migration and invasion in transwell assays in untreated and TGF-β-treated MCF10A cells transiently transfected with vehicle or *CELF1* overexpression construct. (**f**) RNA crosslinking-immunoprecipitation/qRT-PCR of GRE-containing mRNAs from untreated and TGF-β-treated MCF10A cells using two different anti-CELF1 antibodies or mouse IgG. *ACTB* is a non-GRE-containing negative control. (**g**) RNA crosslinking-immunoprecipitation/qRT-PCR of tRFP reporters containing either the wild-type 3′-UTRs for the indicated genes or corresponding mutant 3′-UTRs in which the GRE has been deleted by site-directed mutagenesis. Reporters were immunoprecipitated from untreated and TGF-β-treated MCF10A cells using anti-CELF1 antibody or mouse IgG. (**h**) Immunoblot analysis of indicated epithelial and mesenchymal cell markers and CELF1 in untreated MCF10A cells 72 h post-transient transfection with *CELF1* driven by the CMV Immediate Early promoter in combination with siRNAs targeting Firefly *Luciferase* or each of the ten GRE-containing mRNAs. HSP90 serves as a loading control. (**i**) Quantification of relative cellular migration and invasion in transwell assays in untreated MCF10A cells 72 h post-transient transfection with *CELF1* driven by the CMV Immediate Early promoter in combination with siRNAs targeting Firefly *Luciferase* or each of the ten GRE-containing mRNAs. (**j**) Immunoblot analysis of indicated epithelial and mesenchymal cell markers and CELF1 in untreated MCF10A cells 72 h post-transient transfection with *pL6.3* expression vectors engineered to overexpress Renilla *Luciferase* or each of the indicated individual GRE-containing coding sequences in combination with siRNA-targeting *CELF1*. HSP90 serves as a loading control. (**k**) Quantification of relative cellular migration and invasion in transwell assays (bottom) in untreated MCF10A cells 72 h post-transient transfection with *pL6.3* expression vectors engineered to overexpress Renilla *Luciferase* or each of the indicated individual GRE-containing coding sequences in combination with siRNA-targeting *CELF1*. All panels are representative of a minimum of three experimental replicates. For immunoblots depicted, samples were derived from the same experiment and gels were processed in parallel. In **c,e,i,k**, error bars depict s.d. of the mean. In **f,g** error bars depict s.e.m. *$P \le 0.05$ (Student's *t*-test). See also Supplementary Fig. 3. Full scans of blots are shown in Supplementary Fig. 10.

induced EMT independent of TGF-β treatment (Fig. 4d,e, Supplementary Fig. 3b–d). Again, this was independent of any significant effects on cell proliferation or viability.

To establish a direct physical interaction between CELF1 protein and the GRE-containing EMT drivers, we performed ultraviolet crosslinking/immunoprecipitation/qRT-PCR assays from epithelial and mesenchymal MCF10A cells. Immunoprecipitation from mesenchymal (but not epithelial) MCF10A cells with two distinct anti-CELF1 antibodies revealed significant enrichment of 13 of the 15 endogenous GRE-containing mRNAs tested (Fig. 4f), with no enrichment for RHOB, ATF3 or the non-GRE-containing ACTB mRNA (Fig. 4f).

To determine whether the interaction of CELF1 with the GRE-containing mRNAs was dependent on the GREs within these transcripts, we repeated our experiments using tRFP reporters fused to wild-type 3′-UTRs, or 3′-UTRs in which the GRE had been removed via site-directed mutagenesis (ΔGRE; Fig. 2d). Within TGF-β-treated cells, enrichment of ΔGRE reporters was markedly diminished as compared with wild-type controls (Fig. 4g).

To further establish that CELF1's ability to drive EMT is dependent on its RNA-binding activity, we designed RNA-binding mutants of CELF1 based on previously published structural work implicating five distinct residues distributed through CELF1's three RNA-recognition motifs (RRMs) as mediators of CELF1's RNA-binding functionality[20]. We generated a battery of individual RRM mutants (ΔD1, ΔD2, ΔD3) in which candidate amino-acid residues within each individual RRM were mutated to alanine, and a fourth mutant in which all five residues in the three RRMs were mutated in aggregate (ΔD1–3). Similarly to transfection with wild-type CELF1, cells transfected with the battery of single-RRM CELF1 mutant expression constructs (ΔD1, ΔD2 or ΔD3) underwent EMT independent of TGF-β treatment (Supplementary Fig. 3e). However, transfection of the ΔD1–3 CELF1 construct, harbouring mutations in all three RRMs, failed to induce EMT (Supplementary Fig. 3e).

Immunoprecipitation with anti-FLAG antibody from MCF10A cells transfected with FLAG-tagged wild-type CELF1 resulted in effective enrichment of tested GRE-containing targets. In contrast, no enrichment of GRE-containing mRNAs was obtained from the cells transfected with the FLAG-tagged ΔD1–3 CELF1 mutant (Supplementary Fig. 3f). Cumulatively, these results are consistent with a model in which the CELF1 protein's ability to bind drivers of EMT and increase their relative translation is dependent on the interaction between CELF1's RRMs and the GREs present within the 3′-UTRs of these drivers.

**Genetic ordering of CELF1 and GRE-containing mRNAs.** That CELF1 directly promotes the translation of mRNAs encoding drivers of EMT implies that CELF1 functions genetically upstream of these gene products in the EMT program. To interrogate this model, we co-transfected MCF10A cells with CELF1 along with siRNAs targeting either Firefly luciferase or each of the individual mRNAs encoding the ten GRE-containing genes necessary for EMT (Fig. 3a,b). After 72 h of culture, the transfectants were assessed for EMT. Seven of the ten

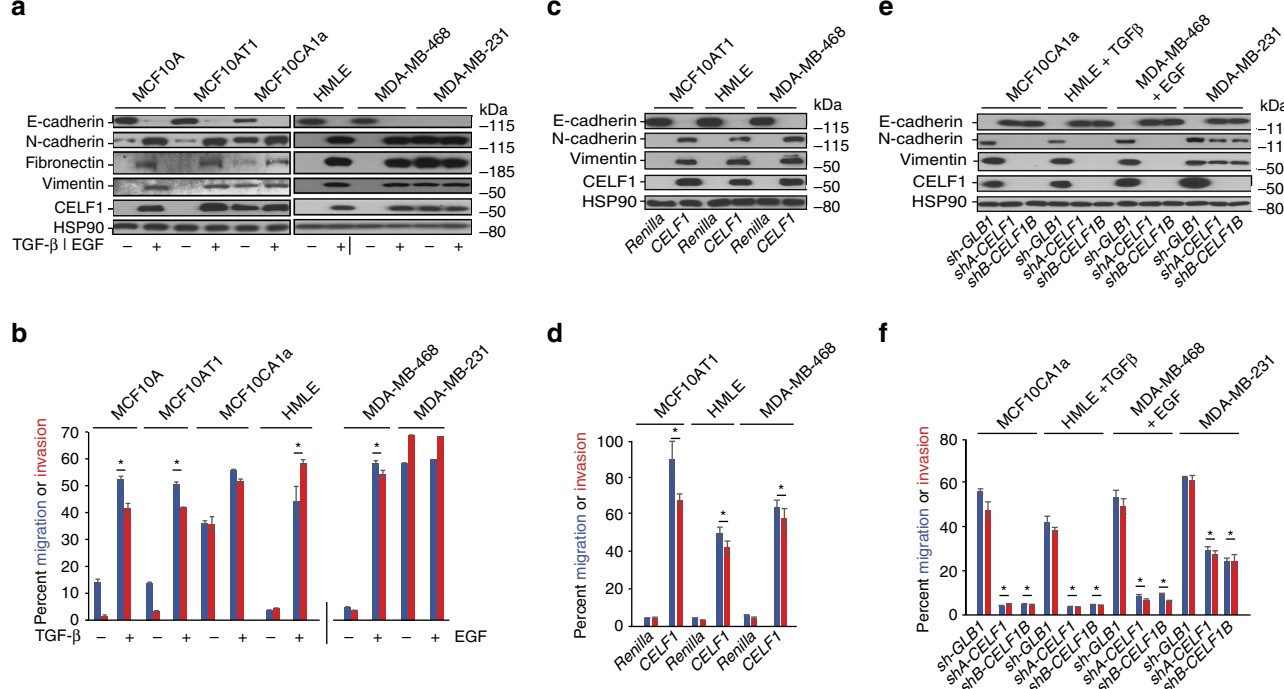

**Figure 5 | CELF1 expression and function are conserved in multiple cellular models of EMT.** (**a**) Immunoblot analysis of indicated epithelial and mesenchymal cell markers and CELF1 in untreated or TGF-β-treated MCF10A, MCF10AT1, MCF10CA1a and HMLE cells, and in untreated or EGF-treated MDA-MB-468 and MDA-MB-231 cells. HSP90 serves as a loading control. (**b**) Quantification of cellular migration and invasion in transwell assays in untreated or TGF-β-treated MCF10A, MCF10AT1, MCF10CA1a and HMLE cells, and in untreated or EGF-treated MDA-MB-468 and MDA-MB-231 cells. (**c,d**) As in **a** and **b**, but in response to stable Renilla Luciferase or CELF1 overexpression in the indicated cell lines, respectively. In **c**, HSP90 serves as a loading control. (**e,f**) As in **a** and **b**, but in response to stable knockdown of β-galactosidase (shGLB1) or two distinct CELF1 shRNAs (shCELF1A, shCELF1B) in the indicated cell lines, respectively. In **e**, HSP90 serves as a loading control. All panels are representative of a minimum of three individual experimental replicates. For immunoblots depicted, samples were derived from the same experiment and gels were processed in parallel. Error bars depict s.d. of the mean. *P ≤ 0.05 (Student's t-test). See also Supplementary Fig. 4. Full scans of blots are shown in Supplementary Fig. 11.

GRE-containing mRNAs tested blocked *CELF1*-induced EMT (Fig. 4h,i, Supplementary Fig. 3g), demonstrating that they function genetically downstream of *CELF1*. Interestingly, disruption of *FOSB*, *JUNB* and *PPARGC1A* did not discernibly impact EMT in cells overexpressing *CELF1* (Fig. 4h,i), indicating that *FOSB*, *JUNB* and *PPARGC1A*, while directly bound by CELF1 and necessary for EMT, are not critical downstream effectors of *CELF1* in the EMT program.

We next transiently overexpressed *Renilla* luciferase (negative control), or each of the six cDNAs sufficient to cause TGF-β-independent EMT in MCF10A cells (Fig. 3c,d) in combination with siRNA-targeting *CELF1*. Consistent with our previous data implicating a function downstream of *CELF1* (Fig. 4h,i), overexpression of *EGR3*, *SEMA6D*, *SNAI1* or *SSBP2* drove EMT even in the context of *CELF1* knockdown (Fig. 4j,k, Supplementary Fig. 3h). In contrast, *CELF1* knockdown abrogated EMT induction by overexpression of *FOSB* or *JUNB* (Fig. 4j,k, Supplementary Fig. 3h). Taken together, these data support a model in which *FOSB* and *JUNB* function upstream of *CELF1* in the same genetic pathway, and the binding of CELF1 to these mRNAs comprises a feed-forward loop within the EMT program to reinforce transition to the mesenchymal state.

**CELF1's role as an EMT inducer is conserved**. To determine whether *CELF1*'s role in the EMT of breast epithelial cells could be further generalized, we first examined two isogenic derivatives of the parental MCF10A line. The non-malignant derivative line, MCF10AT1, is reported to be able to form tumours in immunocompromised mice, but does not metastasize[21]. The malignant derivative line, MCF10CA1a, derived from serial passage of MCF10AT1-derived xenograft tumours, forms aggressive tumours with high metastatic potential in immunocompromised mice[22].

Treatment of the MCF10AT1 line with TGF-β for 72 h resulted in increased expression of CELF1 protein and EMT (Fig. 5a,b, Supplementary Fig. 4a). As in the parental MCF10A line, stable overexpression of *CELF1* promoted EMT in MCF10AT1 cells independent of TGF-β stimulus (Fig. 5c,d, Supplementary Fig. 4b). In contrast to the MCF10AT1 line, the MCF10CA1a cells expressed CELF1 protein *a priori* even in the absence of TGF-β treatment (Fig. 5a). Strikingly, shRNA-mediated knockdown of *CELF1* in MCF10CA1a cells significantly decreased the migratory and invasive potential of these cells, increased the relative expression of E-cadherin, and decreased the relative expression of mesenchymal markers (Fig. 5e,f, Supplementary Fig. 4b).

We next examined the breast epithelial-derived HMLE cell line, which undergoes EMT over a period of 12 days in response to TGF-β (ref. 23). CELF1 protein expression was increased on TGF-β treatment and EMT in HMLE cells (Fig. 5a,b, Supplementary Fig. 4a), mirroring our observations in the MCF10A line. As in the latter line, shRNA-mediated knockdown of *CELF1* blocked EMT of HMLE cells in response to TGF-β (Fig. 5e,f, Supplementary Fig. 4b), while stable overexpression of *CELF1* promoted EMT in the absence of additional stimulus (Fig. 5c,d, Supplementary Fig. 4b).

To determine whether CELF1's role in EMT was constrained to the context of TGF-β signalling, we next turned to the MDA-MB-468 breast epithelial cell line, which undergoes EMT in response to epidermal growth factor (EGF)[24]. Treatment of the MDA-MB-468 line with EGF led to increased expression of CELF1, EMT (Fig. 5a,b, Supplementary Fig. 4a), and increased relative expression of a subset of proteins encoded by the GRE-containing genes identified in our primary MCF10A system (Supplementary Fig. 4c). RNAi-mediated knockdown of *CELF1* blocked EMT of MDA-MB-468 cells, while overexpression of

CELF1 promoted EMT of these cells in the absence of EGF treatment (Fig. 5c–f, Supplementary Fig. 4b).

Finally, we assessed the *CELF1* expression and function in the highly invasive and metastatic MDA-MB-231 breast epithelial cell line, which constitutively expresses CELF1 protein (Fig. 5a,b, Supplementary Fig. 4a,b). RNAi-mediated knockdown of *CELF1* in the MDA-MB-231 line decreased the relative expression of mesenchymal markers, increased the relative expression of E-cadherin, and markedly decreased the migratory and invasive potential of these cells (Fig. 5e,f, Supplementary Fig. 4b). Indeed, steady-state levels of CELF1 protein expression were directly correlated with the metastatic potential of a panel of additional breast cancer cell lines in the absence of stimulus[25] (Supplementary Fig. 4d).

In aggregate, these results indicate that CELF1 both drives EMT and plays a role in maintenance of the mesenchymal state in multiple breast epithelial cell lines and initiating stimuli. It is important to note that manipulation of *CELF1* expression in these experiments did not materially affect proliferation or apoptosis of any of the lines, except for a modest positive effect on proliferation in MCF10AT1 and MDA-MB-468 cells on *CELF1* overexpression (Supplementary Fig. 4e,f).

**CELF1 facilitates *in vivo* lung colonization**. We next hypothesized that misexpression of *CELF1* would impact tumour cell colonization in an *in vivo* xenograft model of experimental metastasis via tail-vein injection. MCF10AT1 cells ectopically expressing *CELF1*, but not *Renilla* luciferase, induced lung metastases in each of the animals tested, developing increased numbers of large macrometastases (mean 49.6 s.d. 7.9 in the OE-*CELF1* group versus a mean of 1.6, s.d. 0.9 in the OE-*Renilla* group; $P = 0.0004$, Student's *t*-test; Fig. 6a,b). Conversely, shRNA-mediated silencing of *CELF1*, but not beta-galactosidase (*GLB1*), attenuated lung metastasis in MCF10CA1a cells (mean 75.4, s.d. 6.4 in the shRNA-*GLB1* group versus a mean of 8, s.d. 3.9 and 12.2, s.d. 2.4 in the shRNA-*CELF1*-A and -B groups, respectively; $P < 0.0001$ in each case, Student's *t*-test; Fig. 6c,d, Supplementary Fig. 5a). We further confirmed the presence or absence of large metastatic foci in the lungs of mice via hematoxylin and eosin staining of histological sections (Fig. 6a,c). The expression of CELF1 and a subset of its regulatory targets was assessed in the lung and was found to be markedly enhanced in the metastatic foci (Supplementary Fig. 5b). *CELF1* can thus drive non-invasive cells to form metastatic lesions, and is critical for the establishment of these lesions by a cell line with *a priori* metastatic potential.

**CELF1 increases with breast cancer progression**. We hypothesized that *CELF1* misexpression might be an underlying feature of human breast cancer. We examined TCGA transcriptional data sets representing 111 cases of human breast cancer and paired normal tissues for changes in the relative expression of the *CELF1* mRNA and its regulatory targets. We note that the expression of classical markers of breast cancer subtypes in our analysis of the TCGA data set (with the exception of PgR and HER2) does not at first glance correspond to commonly presented enrichments since these relative levels are presented in the context of comparison with matched, normal tissue rather than over a broad range of cancer subtypes.

Our analysis revealed similar or decreased levels of GRE-containing mRNA transcripts in several defined molecular subtypes of human breast cancer as compared with normal controls (Fig. 7a). Strikingly, there were essentially no changes in the expression of *CELF1* in this comparison. We thus returned to

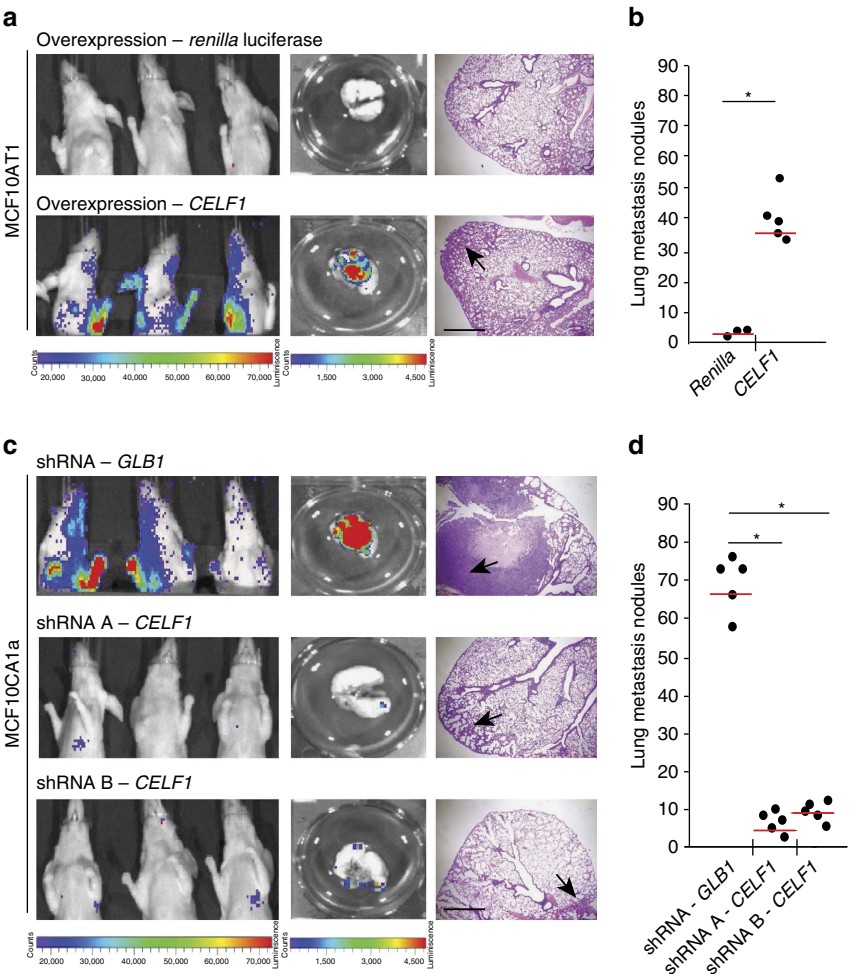

**Figure 6 | CELF1 drives *in vivo* metastatic colonization.** (**a**) The effect of *CELF1* overexpression on metastatic lung colonization was examined. MCF10AT1 cells stably overexpressing either *Renilla* Luciferase or *CELF1* were injected into the tail vein for experimental metastasis within athymic nude mice. The incidence and progression of metastasis was measured by luciferin injection and bioluminescence imaging of *Firefly* Luciferase. Mice were killed on day 100 and metastases were quantified in excised lungs by *ex vivo* bioluminescence imaging. Shown on the right, hematoxylin and eosin staining of the lungs from mice. Scale bar, 1 mm. Black arrows indicate micrometastases. (**b**) The lungs from each group of experimental animals in **a** were surgically excised, fixed overnight in 10% buffered formalin and metastatic nodules were counted. Red line denotes 25th percentile. (**c**) The effect of *CELF1* knockdown on metastasis were examined as in **a**, except that the animals were injected with MFC10CA1a cells stably expressing shRNA hairpins targeting *GLB1* or *CELF1* (2 distinct shRNAs). Scale bar, 1 mm. (**d**) The lungs from each group of experimental animals were surgically excised, fixed overnight in 10% buffered formalin and metastatic nodules were counted. Red line denotes 25th percentile. In **b** and **d**, *$P \leq 0.05$ (Student's *t*-test). See also Supplementary Fig. 5.

our primary MCF10A model, finding that the observed increase in relative CELF1 protein expression associated with the mesenchymal state (Fig. 4a) occurred independently of significant changes in total *CELF1* mRNA expression or ribosomal occupancy (Fig. 7b). However, blocking proteasomal degradation via treatment with MG-132 in epithelial MCF10A cells resulted in marked increases of CELF1 protein in these cells (Fig. 7c). In other systems, the stability and activity of CELF1 protein has been shown to be impacted by phosphorylation[26]. Consistent with these observations, CELF1 protein was characterized by markedly increased phosphorylation of serine and threonine (but not tyrosine) residues in mesenchymal MCF10A cells (Fig. 7d). Under the rationale that *CELF1* gene expression might be similarly regulated in the context of human breast cancer, we next asked whether dysregulation of CELF1 protein expression in breast cancer might be visualized via immunohistochemistry.

We evaluated CELF1 expression in two human breast cancer arrays, together comprising 140 distinct breast cancers and 76 normal adjacent tissue controls. Within this collection, 29 cancers were matched to normal adjacent tissue from the same patients.

Cancerous breast tissue was characterized by more intense and higher proportions of CELF1 staining (mean per cent score 66.7, s.d. 34.7, median score 80 and 35% of tumours were scored at 100%) as compared with non-cancerous tissue (mean per cent score 30.7, s.d. 33.6, median score 10 and 4% of data were scored at 100%; Fig. 7e and Supplementary Fig. 6a,b). Pairwise comparison of matched cancer and normal adjacent control tissues from individual patients revealed a highly significant increase in proportion scores in cancer tissue ($P < 0.0001$, Wilcoxon's signed-rank test; Fig. 7f).

We next correlated relative CELF1 protein expression with a number of different breast cancer patient variables. CELF1 protein expression was not correlated to age ($P = 0.5141$, Kruskal–Wallis test), or to oestrogen receptor, progesterone receptor or HER2 expression ($P$-values = 0.3157, 0.3367 and 0.1633, respectively; Kruskal–Wallis test). However, although overall median proportion scores in breast tumour tissue were already 80%, the proportion of tumour cells positive for CELF1 expression significantly increased as a function of tumour grade (Grade I mean 60.1, s.d. 32.9; Grade II mean 67.2, s.d. 34.5; Grade

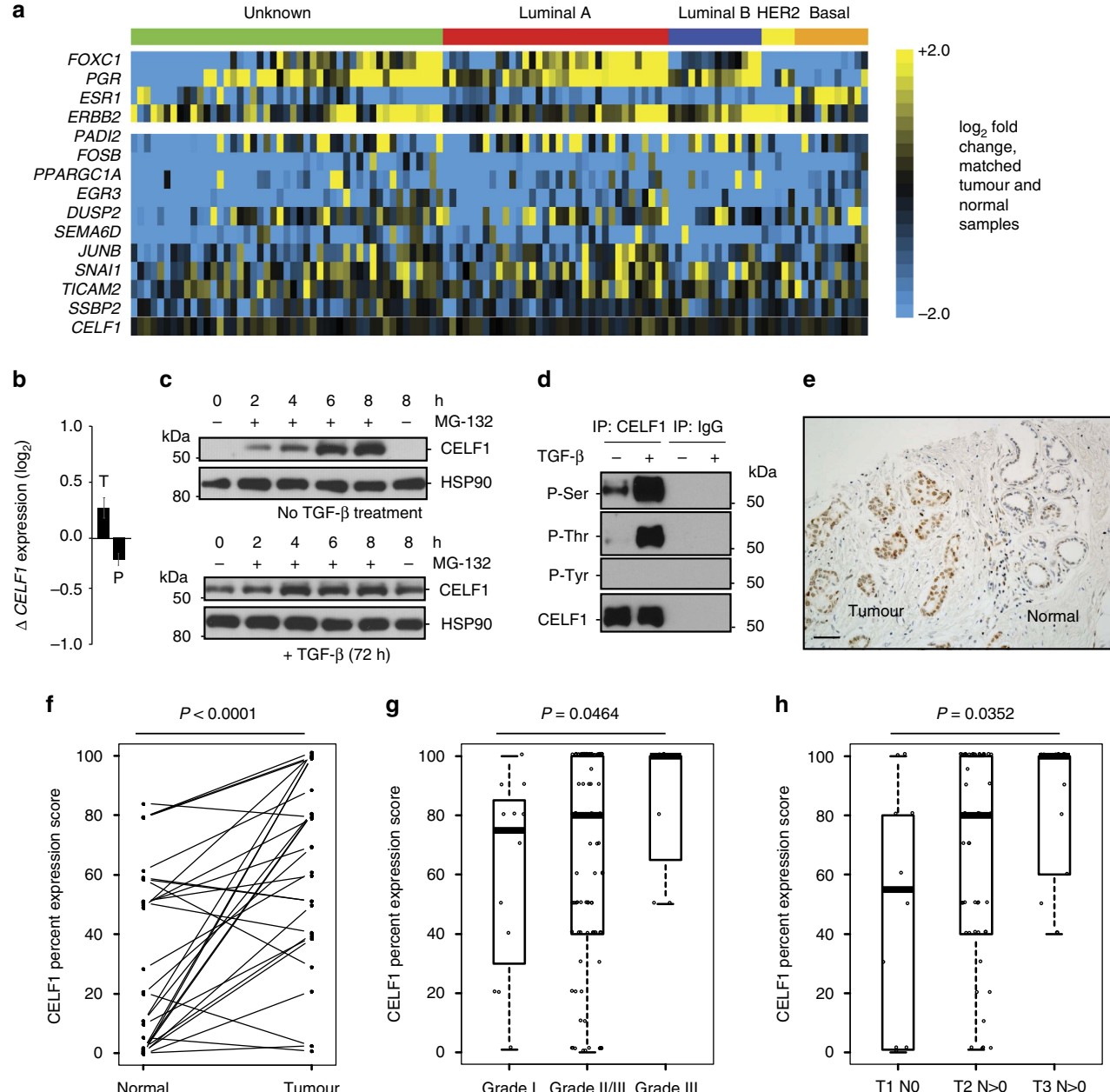

**Figure 7 | Enhanced CELF1 expression in human breast cancer correlates with increased metastasis.** (**a**) Heat map depicting log$_2$-fold difference in mRNA expression of indicated genes between tumour-normal matched pairs for 111 TCGA patients. Topmost coloured bar indicates PAM50 assignments. Columns are ordered by PAM50 subtype. Top: heat map shows exemplary set of genes correlated with subtype, and bottom: heat map depicts genes translationally regulated by CELF1 during EMT and *CELF1*. For each heat map, rows are ordered by decreasing variance from top to bottom. Columns are ordered by PAM50 assignment and within PAM50 assignment by increasing mean log$_2$-fold change of genes from left to right. (**b**) qRT-PCR-based determination of change in steady-state and polyribosomal-bound *CELF1* mRNA using total (*T*) and polyribosomal (*P*) mRNA obtained from untreated and TGF-β-treated MCF10A cells. Data are represented as mean ± s.e.m. All panels are representative of a minimum of three experimental replicates. (**c**) CELF1 is hyperphosphorylated in mesenchymal MCF10A cells. Immunoprecipitates obtained using anti-CELF1 antibody or mouse IgG from MG-132 treated (8 h) MCF10A ± TGF-β (72 h) cells were probed with indicated antibodies. (**d**) Inhibition of proteasome-mediated degradation results in increased CELF1 protein levels in epithelial MCF10A cells. Cells were incubated with MG-132 for the indicated times and immunoblotted for the indicated proteins. HSP90 serves as a loading control. (**e**) Representative image of CELF1 protein expression in mammary carcinoma and tumour adjacent normal mammary gland tissue, as analysed by IHC staining. Image was obtained with × 20 objective. Brown staining indicates antigen, while blue staining represents counterstain. Scale bar, 100 μm. (**f**) Relative CELF1 protein expression as assessed by the per cent score in the 29 matched breast tumour and adjacent normal tissue (*P* < 0.0001; Wilcoxon's signed-rank test). (**g**) Intracellular CELF1 staining was scored in breast tumour samples from patients with tumour size < 20 mm and no lymph node involvement (T1,N0) (*n* = 10), and with lymph node involvement along with tumour size between 20–50 mm (T2,N > 0) (*n* = 53) and > 50 mm (T3,N > 0) (*n* = 13), and the *P*-value of the trend was determined using the Jonckheere–Terpstra test. Box plots represent the 25th to 75th quartiles with the bold horizontal line representing the median value. (**h**) Trend between CELF1 expression and tumour grade I (*n* = 12), grade II–III (*n* = 97) and grade III (*n* = 7). The *P*-value of the trend was determined using the Jonckheere–Terpstra test. Box plots represent the 25th to 75th quartiles with the bold horizontal line representing the median value. See also Supplementary Fig. 6. Full scans of blots are shown in Supplementary Fig. 12.

II–III mean 82.9, s.d. 23.6), revealing an inverse relationship between CELF1 expression level and differentiation state (one-sided $P = 0.0464$, Jonckheere–Terpstra trend test; Fig. 7g). Additional analysis revealed comparatively low proportions of tumour cells staining for CELF protein in smaller early stage tumours without detectable nodal involvement (T1N0 mean 50.2, s.d. 40.3), with progressively increasing rates of positivity in larger tumours that had spread to the lymph nodes (T2N > 0 mean 68.3, s.d. 33.8; T3N > 0 mean 81.5, s.d. 24.8; two-sided $P = 0.0352$, Jonckheere–Terpstra trend test; Fig. 7h), cumulatively indicating that increased CELF1 protein expression is associated with the progression and metastatic spread of breast cancer.

To determine whether CELF1 expression levels had the potential to be of use as a clinical prognostic marker, we performed overall survival analysis of the patients in our data set using the Kaplan–Meier method and a log-rank test. The function form of the CELF1 percentage score indicated an optimal cut-point at 80% positivity. Using this cut-point, higher CELF1 expression was associated with a less favourable outcome, particularly after 5 years of follow-up. However, this association did not achieve significance in the context of the overall comparison (log-rank $P = 0.1963$, LIFETEST procedure).

Finally, to determine whether misexpression of CELF1 protein may occur in other cancer types, we compared CELF1 expression between carcinomas and normal tissue controls of 20 distinct organs on a high-density tissue array. Although increased levels of CELF1 protein expression were noted in carcinomas derived from 15 of the 20 organ types, this was observed in only a low proportion (ranging from 5–20%) of specimens (Supplementary Fig. 6c). These results are consistent with the notion that *CELF1* may contribute to tumourigenesis in other human cancers, but if so less commonly than we describe here in the context of breast cancer.

## Discussion

Regulation at the level of translation can occur via multiple mechanisms including broad effects on the translational machinery via eukaryotic translation initiation factor 2 α-subunit[27]. More limited regulatory programs may be mediated by the binding of distinct *trans*-regulators to common *cis*-regulatory elements present within discrete sets of functionally related mRNA transcripts[28–34]. Here we used a forward and unbiased approach to identify such discrete sets of mRNAs differentially utilized by the translational machinery during EMT. That we elected to focus our search for common *cis*-elements within the 3′-UTR of affected transcripts in this study was a somewhat arbitrary decision. We explicitly note that in our initial reporter validation screen (Fig. 2c) only thirteen of thirty-three 3′-UTRs (39%) derived from polysomally enriched mRNA transcripts conferred regulation of an appreciable magnitude. Beyond this, *in silico* analysis of the 3′-UTRs of polysomally depleted mRNAs in the mesenchymal state did not identify any common motifs within this set. These observations strongly imply that 3′-UTR-independent mechanisms of translational regulation are simultaneously at play in our primary model system, and that additional study of the differentially translated mRNAs that we have identified is likely to reveal additional mechanisms impinging on translational regulation in this context.

That GRE-containing mRNAs comprise a positive regulator class in the context of EMT is at first glance counterintuitive. Previous biochemical and genomic work has consistently demonstrated that GREs promote mRNA destabilization and/or degradation in several different contexts[35]. Consistent with this previous work, comparison of total mRNA levels between epithelial and mesenchymal cells revealed a significant decrease in relative expression of over half of the GRE-containing mRNAs

despite their marked enrichment within mesenchymal polysomal fractions. This observation is not entirely without precedent—a previous study demonstrated that direct tethering of CELF1 protein to a reporter gene resulted in a decrease of relative expression of reporter RNA concomitant with increased relative expression of reporter protein[36]. It would be interesting to determine whether similar increases in protein expression from GRE-containing mRNAs were observed in the context of previous model systems where GRE-mediated destabilization of mRNA has been documented[37].

The data we present with regard to how CELF1 may be differentially regulating translation of the GRE-containing mRNAs is most consistent with a model involving 5′ $^{7mG}$ cap-dependent translational initiation. Work by other groups has provided strong evidence that phosphorylation of eIF4E is required for EMT and metastasis via translational control of a subset of EMT inducers[33,38,39]. It is tempting to speculate that binding of GRE-containing mRNAs by CELF1 may directly impinge on eIF4E-dependent mechanisms.

A subset of the CELF1-regulated GRE-containing mRNAs that we have identified have been previously implicated in the process of EMT[3,40,41] or the metastatic progression of tumours in different contexts[42–47]. *SNAI1*, one of the central and most well-characterized regulators of EMT[3], has been shown to be regulated at the level of transcription[3], post-transcription[8] and post-translation[48]—highlighting that fine tuning of gene expression during EMT is likely to require interactions involving crosstalk among several levels of gene regulation. Our work expands on this theme by revealing post-transcriptional regulation impinging on a group of EMT effectors. To our knowledge, beyond *SNAI1*, none of the mRNAs identified in this study have been shown to be translationally regulated via their 3′-UTRs in EMT. Beyond this, we were indeed surprised that while we observed transcriptional upregulation of *SNAI1* ( + 3.2), *SNAI2* ( + 2.6), *ZEB1* ( + 1.3) and *ZEB2* ( + 0.9) on EMT, only *SNAI1* was polysomally enriched ( + 2.8—all values $\log_2$). Differential expression or polysomal enrichment/depletion for *TWIST1*, *TWIST2* and *FOXC2* in our model system were even less remarkable, and Goosecoid was not detected.

Although *CELF1* is best known for its role in the pathogenesis of Type 1 Myotonic Dystrophy[19], it has been previously shown to play a protective role in chemotherapy-induced apoptosis[49–52] and was identified as a top driver of colorectal tumourigenesis in an unbiased mutagenesis study[53]. However, none of these studies have delved into the mechanism by which *CELF1* effects these various phenotypes. Given the post-translational mode of regulation of CELF1 expression, it is perhaps unsurprising that examination of TCGA data sets revealed neither significant variations in relative *CELF1* mRNA expression among different breast cancer subtypes nor in comparisons of tumour with normal tissue. One strong indicator for metastatic potential of a tumour is the differentiation status of the primary tumour[54], with poorly differentiated tumours tending to be the most metastatic. Our analysis revealed that CELF1 expression was directly correlated to tumour stage and inversely correlated to tumour differentiation status, consistent with the notion that CELF1 expression is correlated to metastatic potential. Although the power of our analysis of CELF1 expression as a prognostic marker with regard to disease outcome was handicapped by the high median value of CELF1 score observed in breast cancer tumour tissues, further investigation in larger annotated tissue sets may solidify this association. It will also be of interest in the future to investigate whether relative CELF1 protein expression is increased further in tumour invasive fronts, which have been shown to be enriched in migratory tumour stem cells[55].

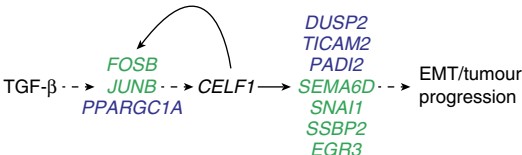

**Figure 8 | Working model illustrating translational regulation at play in TGF-β-mediated EMT of breast epithelial cells.** TGF-β induces CELF1 protein expression, which in turn induces translational upregulation of a cohort of mRNAs that are either necessary (blue) or both necessary and sufficient (green) to induce EMT. The model incorporates results of our genetic ordering analysis (Fig. 4), which suggests *PPARGC1A*, *FOSB* and *JUNB* initiate a feed-forward loop with *CELF1* through an indirect and as-yet-undefined mechanism.

We propose a novel post-transcriptional regulon[56] active in EMT and cancer progression (Fig. 8), adding to the established catalogue of post-transcriptional regulatory events underlying EMT[57–60]. It is important to reiterate that the vast majority of the components of the regulon that we have identified are defined by either no change in or reduction of relative total mRNA expression in publicly available breast cancer transcriptional data sets. Although a full characterization of each of the components of the regulon is beyond the scope of this work, it is reasonable to consider the possibility that in future studies many of CELF1's downstream targets will be found to be similarly misexpressed at only the protein level in primary human breast cancers. Our findings underscore the potential importance of discrete post-transcriptional regulons in the context of fundamental processes underlying tumourigenesis and cancer progression.

## Methods

**Cell culture and treatment.** MCF7, 293T and MCF10A cell lines were obtained from the ATCC (Manassas, VA, USA), whereas the MCF10CA1a.cl1 (or MCF10CA1a) cell line was obtained from the Karmanos Cancer Institute, Detroit, MI, USA) and cultured as described previously[17,22]. MCF10AT1, HMLE, MDA-MB-468 and MDA-MB-231 cell lines were kind gifts from Drs Jason I. Herschkowitz (University at Albany, Rensselaer, NY, USA), Michael J. Toneff and Jeffrey M. Rosen (Baylor College of Medicine, Houston, TX, USA), Bingliang Fang (The University of Texas, MD Anderson Cancer Center, Houston, TX, USA) and Laura Camacho (Baylor College of Medicine), respectively, and cultured as described previously[21,24,25]. MCF10A, MCF10AT1 and MCF10CA1a cells were treated for 3 days and HMLE cells for 12 days with TGF-β1 (5 ng ml$^{-1}$; R&D Systems, Minneapolis, MN, USA). MDA-MB-468 and MDA-MB-231 cells were serum-starved (0.5% fetal bovine serum) for 24 h and then treated with EGF (50 ng ml$^{-1}$) for 3 days. For the proteasomal inhibition, cells were treated with 20 μM of MG-132 (Enzo Life Sciences, Farmingdale, NY, USA) for the indicated times.

**Polysomal profiling.** MCF7 and MCF10A cells ± TGF-β1 for 72 h were treated for 30 min at 37 °C with either 100 μg ml$^{-1}$ cycloheximide (Sigma-Aldrich, St Louis, MO, USA) or 100 μg ml$^{-1}$ puromycin (Sigma-Aldrich). Post-treatment, cells were washed with cycloheximide or puromycin containing phosphate-buffered saline before being lysed using polysome lysis buffer (10 mM Tris-Cl, pH 7.4, 5 mM MgCl$_2$, 100 mM KCl, 1% (v/v) Triton X-100, 0.5% (w/v) deoxycholate in RNase-free water, supplemented with protease/phosphatase inhibitor cocktail, 2 mM dithiothreitol, 1,000 U ml$^{-1}$ RNasin and 100 μg ml$^{-1}$ cycloheximide) for 10 min on ice. Post-nuclear fractions were obtained by centrifuging the lysates at 16,000$g$ for 15 min at 4 °C. One-tenth (v/v) of the lysate was reserved for total RNA isolation. Fifty optical density units of lysate was layered on a 10–50% sucrose gradient and spun for 4 h at 100,000$g$ and 4 °C. Post-centrifugation, fractions were obtained using a BR-184 tube piercer and syringe pump (Brandel, Gaithersburg, MD, USA). Absorbance at 254 nm was monitored using a UA-6 UV detector (Teledyne ISCO, Lincoln, NE, USA) and a DI-158U USB data acquisition device (DATQ Instruments, Akron, OH, USA). Peak Chart Data Acquisition Software was used to process and represent the data as absorbance at 254 nm versus time of fractionation.

**RNA and protein isolation from polysomal fractions.** TRIzol LS reagent (Life Technologies, Carlsbad, CA, USA) was used to extract RNA from the various polysome fraction and total lysate aliquots as per the manufacturer's recommendations. Total RNA was resolved on a 1% denaturing agarose gel to verify the progressive distribution of the 5S, 18S and 28S RNA among the different fractions (uncropped scans of gel is provided in Supplementary Fig. 7c). For

protein isolation, trichloroacetic acid was added at a final concentration of 20% to aliquots of the different polyribosomal fractions. These aliquots were incubated on ice for 30 min, and then centrifuged at 21,000$g$ for 30 min. The pellets were washed twice with 500 μl cold acetone and centrifuged for 15 min at 21,000$g$. Pellets were air dried and then resuspended in Laemmlli buffer and resolved by sSDS-polyacrylamide gel electrophoresis. Blots were probed using eIF3C, rSP6 and PABP antibodies (kind gifts from Dr Richard Lloyd, Baylor College of Medicine).

**Next-generation RNA sequencing and data analyses.** Triplicates of poly-ribosomal enriched and total mRNA (2 μg each) from control and TGF-β-treated MCF7 and MCF10A cells were spiked with ERCC RNA Spike-in Mix 1 or Spike-in Mix 2. Poly(A) RNA was isolated using Dynabeads mRNA DIRECT Micro Kit (Life Technologies) according to manufacturer's instructions. The Ion Total RNA-Seq Kit v2 (Life Technologies) was used according to manufacturer's recommendations to prepare representative barcoded cDNA libraries for strand-specific RNA sequencing. Following assessment of yield and size distribution of the amplified cDNA, the Ion PI Template OT2 200 Kit v2 (Life Technologies) was used as per manufacturer's instructions to generate tem-plate Ion PI Ion Sphere particles using the Ion OneTouch 2 System. Ion PI Chip preparation was done using the Ion PI Sequencing 200 Kit v2 (Life Technologies) and sequenced on the Ion Proton Sequencer platform. Raw sequence data were aligned to the latest draft of the human genome using the Tophat2 and Bowtie2 algorithms within the Partek Flow Suite. Using Partek's Gene Specific Analysis feature, we calculated enrichment or depletion of polyribosome-associated mRNA in each fraction relative to total cellular mRNA. We then plotted the data in terms of mesenchymal polyribosomal content over epithelial polyribosomal content in both cell lines.

***In silico* analysis to identify regulatory elements.** Any number of repetitive motif occurrences ($\geq 6$ width $\leq 10$) in the 3′-UTRs of the putative positive and negative regulators were queried using two *de novo* pattern-finding algorithms, the MEME Suite[15] and BioProspector[16].

**Construction of expression reporters and pBUTR screening.** Donor plasmids with 3′-UTRs corresponding to mRNAs in the enrichment/depletion analysis were generated by PCR amplification of UTR elements from MCF10A genomic DNA using primers containing the Gateway *att*B2r and *att*B4 recombination sequences (Supplementary Table 1). These UTRs were individually 'recombineered' into a piggyBac-based bi-fluorescent reporter system (pBUTR) we have previously described to serve as the 3′-UTR of tRFP within this system[17]. GREs in five of the pBUTR reporters were deleted via site-directed mutagenesis using QuickChange XL Site-Directed Mutagenesis Kit (Agilent Technologies, La Jolla, CA, USA) and primers listed in Supplementary Table 1. Transfection and post-transfection selection with G418 (Teknova, Hollister, CA, USA) of pBUTRs in MCF10A cells were done as described previously[17]. Following selection, cells were treated with TGF-β for 72 h and then analysed by flow cytometry. Expression of tGFP, tRFP and E-Cadherin were determined by flow cytometry using a FACSCalibur system (BD Biosciences, Franklin Lakes, NJ, USA)[17]. Data were analysed using FlowJo version 9 (Tree Star, Ashland, OR, USA). The median fluorescence intensity of tRFP expression were normalized to tGFP expression in each pBUTR comparison between untreated and TGF-β-treated MCF10A cells and then relative fold increase or decrease in tRFP expression was determined for each reporter.

**RNA extraction and qRT-PCR.** RNA was isolated using TRIzol (Life Technologies) as per the manufacturer's recommendations. In all, 1 μg of total RNA was reverse-transcribed using SuperScript III Reverse Transcriptase (Life Technologies) primed with random hexamers. Control reactions were set up without any reverse transcriptase. The cDNAs were subsequently used for qRT-PCR reactions using KAPA SYBR FAST Universal 2× qPCR Master Mix (KAPA BIOSYSTEMS, Wilmington, MA, USA) and indicated primers (Supplementary Table 1). Data were normalized to TBP expression and analysed by the $-\Delta\Delta Ct$ method.

**Plasmid constructs.** All coding sequence entry constructs for overexpression other than *PADI2* and *PPARGC1A* were kind gifts from Dr Kenneth L. Scott (Baylor College of Medicine). *PADI2* and *PPARGC1A* cDNAs were obtained from Thermo Scientific (Pittsburgh, PA, USA) and Addgene (Cambridge, MA, USA), respectively. pEGFP-N1-CELF1 was a kind gift from Dr Lubov Timchenko (Baylor College of Medicine)[61]. p3XFLAG-CELF1 was a kind gift from Dr N. Muge Kuyumcu-Martinez (The University of Texas Medical Branch at Galveston)[26]. Individual RNA-binding mutants of *CELF1* were generated via site-directed mutagenesis using QuickChange II XL Site-Directed Mutagenesis Kit (Agilent Technologies) and primers listed in Supplementary Table 1. The combined RNA-binding mutant of *CELF1* was generated by generating the mutants in six overlapping fragments and then combining the fragments in a single-tube isothermal reaction using the Gibson Assembly Master Mix (New England Biolabs, Ipswich, MA, USA). RNAi-mediated inhibition was performed using Silencer Select siRNAs (Life Technologies; Supplementary Table 2). The *MBNL1*, *MBNL2*,

CELF1, CELF2 and Firefly *Luciferase* siRNAs were kind gifts from Dr Thomas A. Cooper (Baylor College of Medicine; Supplementary Table 2).

**Luciferase reporter constructs and luciferase assay.** The 3′-UTR reporters were constructed by amplifying the endogenous 3′-UTR from MCF10A genomic DNA. *Xba*I and *Apa*I sites were added to the 5′ and 3′ ends of the fragment during the PCR reaction to facilitate subcloning into the *Xba*I and *Apa*I sites of the Renilla *Luciferase* vector (pRL-TK CXCR4 6x) (ref. 18). The pFR-EMCV (TK-driven firefly and IRES-driven Renilla and 3′-UTR) were subcloned by amplifying the IRES sequence from *pINDUCER10* (a kind gift from Dr Thomas Westbrook), and subcloning this into the *Nhe*I and *Nde*I sites on the pFR_HCV_xb vector (ref. 62). The 3′-UTRs were subcloned into the pFR-EMCV by the *Xba*I and *Not*I sites. Luciferase assays were performed using the Dual-luciferase reporter assay system (Promega, Madison, WI, USA) as per the manufacturer's protocol on a Tecan M200 multimode reader using Tecan Magellan software (Tecan).

**Transfection and transduction.** Transient transfection was performed using Lipofectamine LTX (Life Technologies), per the manufacturer's instructions. Silencer Select siRNAs were used at 100 nM final concentration. *pGIPZ* lentiviral particles were generated by transfection of 293Ts using Mirus TransIT-293T (Mirus Bio LLC, Madison, WI, USA), per the manufacturer's instructions. The pLenti6.3 lentiviral particles were generated by transfection of 293Ts as described previously[63]. For transduction, early passage cells were seeded at 500,000 cells per 10-cm$^2$ dish 1 day before infection and transduction was performed as described previously[64]. Transductants were selected with Puromycin (2 µg ml$^{-1}$) and Blasticidin (5 µg ml$^{-1}$) for *pGIPZ* and *pL6.3*, respectively. In all cases, gene silencing or ectopic overexpression was verified by immunoblotting.

**Cell lysis and western blot.** Cell lysis and western blot was performed as described previously[17]. Supplementary Table 3 provides the list and associated information of antibodies used in the current study. The CELF2, MBNL1 and MBNL2 antibodies were kind gifts from Dr Thomas A. Cooper (Baylor College of Medicine). All blots were subsequently stripped, and re-probed for HSP90 to confirm equal loading. Uncropped scans of all western blots are provided in Supplementary Figs 7–12.

**Cell proliferation and viability assays.** Cell proliferation was quantitated using a mitochondrial colorimetric assay (MTT assay, Sigma-Aldrich) as per the manufacturer's recommendations. The absorbance was measured at 570 nm and post-measurement corrected by subtracting absorbance at the reference wavelength of 690 nm. The results, expressed as relative optical density, were obtained for three different experiments and expressed as mean ± s.d. Cell viability was quantified by flow cytometry analysis of cells following staining with annexin V-fluorescein isothiocyanate (BD Biosciences) and propidium iodide (Life Technologies). Cells treated with bleach (0.1%, v/v) for 10 min were used as positive control for annexin V/PI staining.

**The cancer genome atlas analysis.** The TCGA data portal (tcga-data.nci.nih. gov/tcga/) was used to download Breast invasive carcinoma RNASeqV2 normalized gene-expression data on 111 tumour–normal matched pairs. PAM50 assignments were downloaded (tcga-data.nci.nih.gov/docs/publications/brca_2012/BRCA.547.-PAM50.SigClust.Subtypes.txt) as delineated in ref. 65. Samples without PAM50 assignment were classified as 'unknown'. Statistical analysis was performed using the R statistical software. Tumour-normal log$_2$-fold changes were calculated using the following equation: $FC_{ij} = \log_2 [(T_{ij}+1)/(N_{ij}+1)]$, where $FC_{ij}$ represents the fold change for patient $i$ in gene $j$. $T_{ij}$ and $N_{ij}$ represent the normalized read counts for gene $j$ in tumour and matched normal samples of patient $i$, respectively.

**Genetic ordering experiments.** For genetic ordering experiments untreated MCF10A cells were co-transfected with CMV IE promoter driven *CELF1* and one of two distinct *Silencer* Select siRNAs targeting each of the ten GRE-containing mRNAs. For reverse epistasis, we transiently co-transfected MCF10A cells with *pLenti6.3* backbone vector plasmids (not virus) designed to overexpress each of the indicated coding sequences and *pGIPZ* backbone *CELF1* or β-galactosidase (*GLB1*) shRNA plasmids. For both epistasis and reverse epistasis, 72 h following transfection cells were assayed for EMT via expression of canonical molecular markers, and in the context of transwell migration and invasion experiments as described above.

**Animal studies.** All mouse procedures were approved by the Institutional Animal Care and Use Committees of the Baylor College of Medicine. Six-week-old spontaneous mutant T-cell-deficient female homozygous nude mice (NCr-*Foxn1*$^{nu}$) (Taconic, Hudson, NY, USA) were used for the experimental metastasis experiments ($n = 5$ in each experimental group).

To assess the experimental metastatic potential of cells, 10$^6$ MCF10CA1a-*GLB1* shRNA, MCF10CA1a-*CELF1*-A shRNA, MCF10CA1a-*CELF1*-B shRNA, MCF10AT1-*Renilla* overexpressing, and MCF10AT1-*CELF1* overexpressing cells labelled with GFP-Firefly luciferase were injected into 5 animals per cell type via

the tail vein. Mice were assessed weekly for metastasis using *in vivo* bioluminescence imaging using an IVIS Imaging System (IVIS imaging system 200, Xenogen Corporation, PerkinElmer, Waltham, MA, USA). Mice were killed on day 100 post tail-vein injection at which time the lungs were surgically removed and fixed using 10% neutral buffered formalin and the number of lung tumour nodules was counted using a dissection microscope. Lungs were further subjected to hematoxylin and eosin staining and immunohistochemistry using anti-CUGBP1 antibody [3B1] (ab9549; Abcam, Cambridge, MA, USA) (1:500), DUSP2 (PA5-28775; Thermo Scientific, Rockford, IL, USA) (1:100), JUNB [C37F9] (3195; Cell Signaling, Beverly, MA, USA) (1:400), Snail1 (bs1371-R; Bioss, Woburn, MA, USA) (1:50), and SSBP2 (LS-B5585; LifeSpan Biosciences, Seattle, WA, USA) (1:400). Images were obtained using Axio Zoom.V16 microscope (stereo) at × 40 objective.

**RNA immunoprecipitation and data analyses.** In all, 10$^7$ untreated or TGF-β-treated (72 h) MCF10A cells were placed on ice, irradiated once with 150 mJ cm$^{-2}$ at 254 nm using a UV Crosslinker (Spectroline, Westbury, NY, USA) and lysed for 15 min on ice in lysis buffer (100 mM KCl, 5 mM MgCl$_2$, 10 mM HEPES, pH 7.0, 0.5% Nonidet P-40 detergent supplemented with fresh 1 mM dithiothreitol, 1,000 units ml$^{-1}$ RNAsin (Promega), and Mini protease inhibitor cocktail (Roche Diagnostics, Indianapolis, IN, USA)). The post-nuclear cytosolic fraction was collected by spinning the cells at 16,000$g$ for 10 min at 4 °C and 10% (v/v) was removed for input sample. In all, 4 mg of lysates were subjected to immunoprecipitation using 10 µg of two CELF1 antibodies (Clone 3B1, EMD Millipore, Billerica, MA, USA and Clone 1.T.9, Santa Cruz Biotechnology, Dallas, TX, USA) or 10 µg of mouse IgG using Pierce Crosslink IP Kit (Thermo Scientific). Aliquots were sequestered for confirming successful immunoprecipitation through SDS-polyacrylamide gel electrophoresis. The rest of the immunoprecipitated complex was digested with 30 µg of proteinase K to release the ribonucleoprotein complex. TRIzol LS reagent (Life Technologies) was subsequently used to extract RNA from the immunoprecipitates and the input samples following manufacturer's recommendations.

The RNAs isolated above were used for first stand cDNA using SuperScript III Reverse Transcriptase (Life Technologies) and random hexamers. Control reactions were set up without any reverse transcriptase. The cDNAs were subsequently used for qRT-PCR reactions using KAPA SYBR FAST Universal 2 × qPCR Master Mix (KAPA BIOSYSTEMS) and indicated gene-specific primers (for oligonucleotide sequences refer to Supplementary Table 1). Each RNA immunoprecipitation (RIP) RNA fractions' threshold cycle ($C_t$) value was normalized to the input RNA fraction $C_t$ value ($\Delta C_t$) to account for RNA sample preparation differences using the following formula: $\Delta Ct$ (normalized (RIP)) = (Ct (RIP) − (Ct (Input) − Log2 (Input dilution factor))). The 'per cent input' for each RIP fraction was calculated as the linear conversion of the normalized RIP $\Delta Ct$ using the formula: % Input = $100 \times 2^{(-\Delta Ct \text{ (normalized (RIP))})}$. The normalized RIP fraction $C_t$ value were then adjusted for the normalized background (NS (non-specific) Ab) fraction $C_t$ value (first $\Delta\Delta Ct$) using the formula: $\Delta\Delta Ct$ (RIP/NS) = $\Delta Ct$ (normalized RIP) − $\Delta Ct$ (normalized NS). Finally, the immunoprecipitation 'fold enrichment' above the sample specific background (linear conversion of the first $\Delta\Delta Ct$) was calculated using the formula: Fold enrichment = $2^{(-\Delta\Delta Ct \text{ (RIP/NS)})}$.

For RNA-IP comparing the impact of GRE mutants, the exact same protocol was followed. A tRFP forward and 3′-UTR-specific reverse primers were used for qRT-PCR, making the detections reporter-specific.

***In vitro* transwell migration and invasion assays.** *In vitro* migration and invasion assays were either performed in Transwell inserts (8 µm pore size; Corning Incorporated, Corning, NY, USA), uncoated or coated with Basement Membrane Extract (BD Biosciences, Bedford, MA, USA) or using Culturex 96-Well Cell Migration and Invasion Assay kits (Trevigen, Gaithersburg, MD, USA). In each case, cells were serum-starved overnight, treated with 10 µg ml$^{-1}$ Mitomycin-C for 2 h, trypsinized and introduced into the upper chamber ($5 \times 10^4$). The chemoattractant in the lower chamber was culture medium supplemented with 5% horse serum. Incubation time was 16 h for all cells, except for MDA-MB-231, in which case incubation time was 4 h. Following incubation, the migratory and invasive cells were either stained with crystal violet following fixation and imaged using an inverted microscope or quantified by Calcein-AM as per manufacturer's recommendations. Data obtained were used to analyse per cent migration and invasion and were expressed as mean ± s.d. Percentages reflect data from the Culturex 96-Well Cell Migration and Invasion Assay kits, and crystal violet staining depicts qualitative analysis.

**Tissue microarray analysis.** Breast cancer tissue microarray containing tissue samples from 150 breast cancer patients (HBre-Duc150Sur-01), breast tissue array containing 90 normal adjacent breast tissue samples (HBre-Duc090Sur-01), and high-density multiple organ tumour tissue array with normal tissue as control (MC5003b) were obtained from US Biomax (Rockville, MD, USA). The TMAs were stained for CELF1 expression using routine methodology using the trypsin enzymatic antigen retrieval solution (Abcam) and the anti-CUGBP1 antibody [3B1] (ab9549; Abcam) at a 1:500 dilution. The stained slides were scored by a

pathologist (D.G.R.) blinded to the identity of the tissue cores as *per cent* of CELF1-positive cells with a range of 0 to 100). Tissue cores representing stromal (non-ductal) tissue were excluded from analysis. For the tumour-normal pairs the scores in both samples and their changes (tumour-normal) were summarized.

**Statistical analysis.** Laboratory data are presented as mean ± s.e.m. except otherwise stated. When two groups were compared, the Student's *t*-test was used unless otherwise indicated and a $P < 0.05$ was considered significant. In human samples, CELF1 per cent scores were summarized using frequency, median and histogram plots. Wilcoxon's signed-rank test was used to compare the CELF1 protein expression between normal and tumour tissues. Spearman's correlation and non-parametric tests, inclusive of Wilcoxon's rank-sum test and Kruskal–Wallis test, were used to examine the association between CELF1 protein expression with available patients' characteristic factors and other markers. For determining trends of breast cancer progression in terms of CELF1 expression the Jonckheere–Terpstra proportion trend test[66,67] was used.

**Data availability.** The RNA sequencing data generated in this work have been deposited in NCBI's Gene Expression Omnibus and are accessible through GEO Series accession number GSE81955 (https://www.ncbi.nlm.nih.gov/geo/query/acc.cgi?acc=GSE81955). The TCGA data referenced during the study are available in a public repository from the TCGA website (https://tcga-data.nci.nih.gov/docs/publications/tcga/). All other data supporting the findings of this study are either included in the manuscript or available on request from the corresponding author.

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

## Acknowledgements

This research was supported by the Nancy Chang Foundation, an American Cancer Society—Athena Water Breast Cancer Research Scholar Grant (RSG-15-088-01RMC and NCI CA190467) to J.R.N., the Baylor College of Medicine Comprehensive Cancer Training Program enabled by RP101499 from the Cancer Prevention and Research Institute of Texas to A.C. Use of the Cytometry and Cell Sorting cores at Baylor College of Medicine was subsidized by additional funding from the NIH (AI036211, CA125123 and RR024574). We thank Drs R. Singh, J.I. Herschkowitz, M.J. Toneff, B. Fang, L. Camacho, K.L. Scott, T.A. Cooper and G. Goodall for sharing cell lines and reagents, Dr J.H. Bayle for help with microscopy, Dr S. Zhang and X. Gao for generous access to the IVIS imaging system, and Dr C. Gutierrez, Dr C.S. Nagi and C. Chenault for access to the resources within the Lester and Sue Smith Breast Center. We thank Drs T.A. Cooper, M. Winslow, J.M. Rosen, K.L. Scott, J.D. Wythe and R.A. Poché for critical reading of the manuscript. We also thank R.K. Bryd and S. Boyd for help with the mice xenograft injections.

## Author contributions

A.C. and J.R.N. conceived, designed, performed and analysed experiments, made figures and wrote the manuscript. S.C., J.M.F., N.K. and G.L. contributed to pBUTR and plasmid constructs used in this study. L.M.S. and C.A.S. performed the TCGA analysis and generated the corresponding figures. S.M. contributed to the immunohistochemistry staining. M.M.I. and D.G.R. evaluated and scored the TMA slides. T.W. and S.G.H. performed the statistical analysis on the TMA data. All authors read and approved the final draft of the manuscript.

## Additional information

**Competing financial interests:** The authors declare no competing financial interests.

