## [Peer Review File · Nature Communications]

Reviewer #1 (Remarks to the Author)

In the manuscript entitled, "CELF1 is a central node in post-transcriptional regulation programs underlying EMT" Chaudhury et al. present intriguing new findings into a previously uncharacterized translation control program that is activated during EMT in breast cancer cells. Importantly, although it has been well characterized that significant changes in transcription occur during EMT, few studies have attempted to investigate how gene expression is regulated at the translational level during this process. This is a particularly important question to address, as there is now a growing realization that post-transcriptional control of gene expression has a causative role in disease pathogenesis, particularly cancer. Employing an unbiased approach and various reporter systems the authors identify a subset of mRNAs that are specifically modulated during EMT by the RNA binding protein CELF1. CELF1 has previously been implicated in modulating gene expression of mRNAs harboring cis-regulator elements known as GU-rich elements (GRE). However, the precise mechanism by which CELF1 regulates the expression of these genes remains somewhat controversial, with the majority of studies demonstrating effects on mRNA stability. In this manuscript, the authors identify that CELF1 modulates the translation of GRE-containing transcripts independent to changes in mRNA levels during EMT. However, providing some mechanistic insights of how CELF1 may enhance translation of its targets would strengthen the impact of the manuscript. Interestingly, the authors identify a role for CELF1 and its target genes in promoting EMT, determine that CELF1 drives non-invasive breast cancer cell lines to form metastatic lesions in vivo, and demonstrate that CELF1 protein levels are significantly increased in primary breast cancer specimens. These are very significant findings however some questions regarding the regulation of CELF1 as well as oncogenic activity should be addressed before publication. Overall, this manuscript is timely, will appeal to a broad audience, and upon revision should be considered for publication in Nature Communications.

Major points:

- 1) Chaudhury et al. provide compelling data that a subset of GRE-containing mRNAs is upregulated by CELF1 upon EMT specifically at the translational level. Importantly, the authors demonstrate enhanced binding of CELF1 to this subset of mRNAs during EMT however they fail to further elucidate how enhanced binding of CELF1 promotes translation (as demonstrated by enhanced polysome associated and increased protein levels). The authors should address whether the GRE in target genes enhances translation initiation through binding of distinct initiation factors such as components of eIF4F complex.
- 2) Using a bi-fluorescent reporter system the authors demonstrate that the GRE within the 3' UTRs of target genes can modulate translation of the reporter. However, as CELF1 is known to affect mRNA stability of GRE-containing genes, the authors should assess transcript levels of reporters with and without the GRE (relating to Figure 2C and 3D). Similarly, the authors should also provide data to support their claim that changes in protein levels of target genes "occurred independently of significant positive changes in total mRNA expression as assessed by quantitative RT-PCR analysis (Fig. 2d)". Currently, this data is not provided in Figure 2D.
- 3) It is recommended that Figure 4, in which the authors repeat their findings in additional breast cell lines, be incorporated into the supplementary materials section, as currently inclusion of this figure does not alter the overall message of the manuscript.
- 4) In order to prove that the EMT phenotype mediated by CELF1 is dependent on its bindings to the GRE of select targets, the authors should perform an experiment similar to that presented in Figure 3F by expressing full length in vitro mRNA of a target gene with and without the GRE. Likewise, Chaudhury et al. should also assess the effect of expressing a CELF1 mutant that inhibits RNA binding to GRE-containing mRNAs on EMT.

5) In Figure 5, the authors demonstrate that overexpression of CELF1 promotes non-invasive breast cancer cell lines to form lung metastasis following tail vein injection in nude mice. The authors should also assess protein levels of GRE-containing targets in the resultant lung metastasis.

6) One of the most intriguing findings from this study is that CELF1 protein but not mRNA is increased during EMT and in breast cancer specimens. The authors speculate that this occurs at the post-translational level however it would significantly increase the overall impact of the manuscript if the authors could provide additional evidence that CELF1 is regulated at the post-translation level during EMT and in cancer specimens by assessing for example CELF1 phosphorylation status. For example they could simply perform a Western Blot with Phospho-specific antibodies, which is commercially available.

Minor Points:

1) Upon identification of mRNAs translationally enhanced during EMT, the authors specifically performed analysis on the 3' UTR of these genes to determine whether they harbor a common cis-regulatory element. However, the authors should comment on why they focus on the 3'UTR and also expand on a possible contribution of 5' UTRs towards mRNA translation.

2) The authors employ wild-type and mutant ZEB1 3' UTRs as controls in experiments presented in Figure 2C. However, they should provide a brief explanation for the reader as to why wild-type and mutant ZEB1 3' UTRs serve as appropriate controls in this reporter system.

3) For consistency, the authors should refer to constructs harboring a deletion of the GRE as either GRE mutant or Δ GRE throughout the manuscript.

4) From the unbiased polysome profiling analysis performed in Figure 1 and 2 Chaudhury et al. also identify a polysome depletion class of mRNAs upon EMT. The authors should comment upon this and propose potential mechanisms of translational regulation that may exist during EMT in the discussion section.

5) On page 6, "UTRS" should be changed to "UTRs".

Reviewer #2 (Remarks to the Author)

The manuscript is well presented and the study provided convincing evidence for the translational regulation of EMT using an unbiased approach and highlights the need to consider more than transcriptional approaches to defining EMT. It broadens the recognized role of CELF1, and defines a genetic ordering including a feed forward component that remains to be further characterized. The regulation of CELF1 also remains to be investigated, and this is important in finding ways to gauge the extent to which the CELF1-associated regulon is active in a given scenario.

The methodology appears sound, the study is innovative and novel, and the analytical approaches appear robust.

Major Points:

1. In cross-comparing the MCF10A and MCF7 systems, the authors "utilized the rationale that any common regulatory events observed in both models would reflect a regulatory response to TGF- β signaling rather than an event associated with EMT per se." However, it is not known where the EMT blockade exists? It could be downstream of polyribosomal loading / protein synthesis?

2. Snail was found to be present amongst the 72 putative positive regulators. It seems likely that other recognizable EMT drivers should have been present?
3. The expression of classical markers of breast cancer subtypes does not seem to correspond to expected enrichment, with the exception of PgR and HER2? For example, ESR1 is enriched in basal subgroup, which should be classically ER-negative. Is this because it is expressed compared to normal rather than empirically? It may warrant a comment? Nonetheless the point about CELF1 and the translational targets is made.
4. Other modes of post-transcriptional regulation are mentioned briefly in the introduction but not further discussed; for example the regulatory role of non-coding RNAs including microRNA. Upstream regulatory events may have important roles in this pathway, for instance what underlying mechanisms are involved in CELF1 to bind to GRE elements of the RNA regulome (11 genes)?
5. The number of mRNAs (209) characterized by altered polyribosomal occupancy in this study is not sufficient to claim the study as global.
6. The authors have examined TCGA breast cancer data but not ICGC data. Analysis of ICGC data, which is a much bigger dataset than TCGA, could possibly strengthen findings described in Figure 6, and provide new insights.
7. From the Methods section (Page 17), it appears that the authors have not submitted NGS data to a public domain; It is important to submit not only total RNA NGS but ribosomal profiling data as well.

Minor points:

1. Should the phrase "invasion metastasis cascade" in line 2 of the paper be "invasion and metastasis cascade"?
2. In the following sentence "Within MCF10A cells, the observed increase in relative CELF1 protein expression associated with the mesenchymal state (Figs. 2a, 2b) was independent of both changes in relative expression of total CELF1 mRNA and the ribosomal occupancy of these transcripts (Fig. 6b). This does not seem to correspond to Fig 2a, 2b?"

Reviewer #3 (Remarks to the Author)

This is a dense paper describing translational aspects of the EMT. The authors primarily use MCF-10 and MCF-7 breast cancer lines, with EMT stimulated by TGF- β . This reviewer is not expert in translational regulation and will confine comments to biological relevance. There are probably thousands of papers at this point on the EMT, yet we are still discussing its overall relevance to cancer progression. For a paper to have a significant impact on the field, it must be comprehensive.

In very few instances was EMT induced by anything but TGF- β or tissue culture. One EGF experiment was shown. These data question the generality of the findings. I would have asked if the same set of genes were found upregulated when EMT was induced in multiple cell lines, under multiple conditions.

The other question is the relevance of EMT. The authors show that E-cadherin and other markers change in expression, but there is only one functional metastasis experiment using one of the genes. Motility and invasion data could be easily added. Other investigators view EMT as a marker of stemness, of cellular transdifferentiation, and of plasticity, and it would have been of great

interest to know what happens in these arenas.

The human tumor data has so few Gr III tumors as to be non-informative.

Finally, please do not use the term "master regulator".

Reviewer #1 (Remarks to the Author): Expert in translational control and cancer

In the manuscript entitled, "CELF1 is a central node in post-transcriptional regulation programs underlying EMT" Chaudhury et al. present intriguing new findings into a previously uncharacterized translation control program that is activated during EMT in breast cancer cells. Importantly, although it has been well characterized that significant changes in transcription occur during EMT, few studies have attempted to investigate how gene expression is regulated at the translational level during this process. This is a particularly important question to address, as there is now a growing realization that post-transcriptional control of gene expression has a causative role in disease pathogenesis, particularly cancer. Employing an unbiased approach and various reporter systems the authors identify a subset of mRNAs that are specifically modulated during EMT by the RNA binding protein CELF1. CELF1 has previously been implicated in modulating gene expression of mRNAs harboring cis-regulator elements known as GU-rich elements (GRE). However, the precise mechanism by which CELF1 regulates the expression of these genes remains somewhat controversial, with the majority of studies demonstrating effects on mRNA stability. In this manuscript, the authors identify that CELF1 modulates the translation of GRE-containing transcripts independent to changes in mRNA levels during EMT. However, providing some mechanistic insights of how CELF1 may enhance translation of its targets would strengthen the impact of the manuscript. Interestingly, the authors identify a role for CELF1 and its target genes in promoting EMT, determine that CELF1 drives non-invasive breast cancer cell lines to form metastatic lesions in vivo, and demonstrate that CELF1 protein levels are significantly increased in primary breast cancer specimens. These are very significant findings however some questions regarding the regulation of CELF1 as well as oncogenic activity should be addressed before publication. Overall, this manuscript is timely, will appeal to a broad audience, and upon revision should be considered for publication in Nature Communications.

We are grateful to the reviewer for their positive comments regarding the manuscript, and also for an insightful and tremendously detailed list of points to address that would markedly improve the study. Please find each of these issues addressed in turn below:

Issue 1: Chaudhury et al. provide compelling data that a subset of GRE-containing mRNAs is upregulated by CELF1 upon EMT specifically at the translational level. Importantly, the authors demonstrate enhanced binding of CELF1 to this subset of mRNAs during EMT however they fail to further elucidate how enhanced binding of CELF1 promotes translation (as demonstrated by enhanced polysome associated and increased protein levels). The authors should address whether the GRE in target genes enhances translation initiation through binding of distinct initiation factors such as components of eIF4F complex.

Response: We are intimately familiar with seminal work in regards to the impact that translational initiation factors may have on EMT (e.g. Robichaud et al. *Oncogene* 34:2032-2042 (2015); Smith et al. *Scientific Reports* 5:18233 (2015)), and the contribution of these types of mechanisms to translational control. This topic is certainly being actively investigated in the laboratory. We hypothesize, pending further experimental validation, that enhanced CELF1 binding results in enhanced interaction with phosphorylated eIF4E and thus de-represses the translational block of the GRE-containing mRNAs. But this specific interaction has not been previously described, and in any case this is not the only possible mechanism. Given the admirably incisive and rigorous nature of the bulk of the revisions that the reviewer proposes in this critique, we hope that the reviewer will agree with the

opinion of ourselves and two local, established experts in translational regulation that if this molecular mechanistic work is done properly, these data cannot be comprehensively presented or summarized in a subpanel of this manuscript, a figure of this manuscript, or perhaps even an entire second paper (one of the gentlemen we consulted went so far as to assert that the proposal of these studies would comprise a full-fledged R01 application of three aims). As such, we consider properly performed and genuinely informative studies such as those proposed beyond the scope of this manuscript. We do, however, present and touch upon these potential mechanisms in the discussion.

This being said, to address the reviewer's concerns we now include evidence in the revised manuscript that the translational regulation mediated by CELF1 is via cap-dependent mechanism. This evidence is based on reporter assays comparing the effect of GRE-containing 3' UTRs on a "standard" reporter gene in the context of the broadly used pRL-TK CXCR4 6x reporter plasmid (**Fig. 2f**), or a version of this plasmid in which these UTRs are fused downstream of a second coding sequence whose expression is driven by the EMCV internal ribosome entry site (**Fig. 2g**). Our data indicate that select GRE-containing UTRs do NOT confer differential expression of a reporter gene in the epithelial and mesenchymal states when the expression of this reporter gene is under the control of an IRES, providing evidence that the regulation observed occurs at the level of translational initiation. Within these experiments, relative levels of reporter mRNA were monitored via qRT-PCR, ruling out an mRNA-stability-based mechanism.

Issue 2: Using a bi-fluorescent reporter system the authors demonstrate that the GRE within the 3' UTRs of target genes can modulate translation of the reporter. However, as CELF1 is known to affect mRNA stability of GRE-containing genes, the authors should assess transcript levels of reporters with and without the GRE (relating to Figure 2C and 3D).

Response: This is a critical control and we are embarrassed that we had initially overlooked it. We have determined that the steady state levels of the reporters' transcripts did not differ significantly with and without the GRE (**Fig. 2e**) of the revised manuscript. As described above, we similarly validated the relative levels of expression of our IRES constructs.

Similarly, the authors should also provide data to support their claim that changes in protein levels of target genes "occurred independently of significant positive changes in total mRNA expression as assessed by quantitative RT-PCR analysis (Fig. 2d)". Currently, this data is not provided in Figure 2D.

Response: In regards to this latter point the qRT-PCR data was indeed included in the original Figure 2d. It is likely that the way that we presented this data obscured this fact. We have now re-labeled the figure (**Fig. 2b** in the revised manuscript) to clearly document these measurements.

Issue 3: It is recommended that Figure 4, in which the authors repeat their findings in additional breast cell lines, be incorporated into the supplementary materials section, as currently inclusion of this figure does not alter the overall message of the manuscript.

Response: We sympathize with the reviewer's rationale, but respectfully submit that a demonstration that the pathway that we describe is neither limited to a particular cell type nor a particular EMT-inducing stimulus is a critical part of demonstrating the biological relevance of the model that we present (not only was this consistently demanded in review

of applications for funding of this research, but we would refer the reviewer to Reviewer 3's second point). These data (**Fig. 5** in the revised manuscript) also set up the rationale for using these cell lines for the *in vivo* xenograft experiments represented in the subsequent figure.

Issue 4: In order to prove that the EMT phenotype mediated by CELF1 is dependent on its bindings to the GRE of select targets, the authors should perform an experiment similar to that presented in Figure 3F by expressing full length in vitro mRNA of a target gene with and without the GRE. Likewise, Chaudhury et al. should also assess the effect of expressing a CELF1 mutant that inhibits RNA binding to GRE-containing mRNAs on EMT.

Response: The incredibly elegant former approach would indeed, in theory, be an ideal experiment to support our core model. However, this is a difficult endeavor. From a technical standpoint, we note that although endogenous CELF1 is actively degraded in epithelial MCF10A cells, our ability to overexpress this protein from a synthetic construct to drive EMT indicates that the underlying regulatory mechanisms here are saturable. One might then expect that similar overexpression of any of the GRE-containing mRNAs, either with or without the GRE, would similarly drive EMT, especially given that CELF1 is an inducer and not repressor in this context. We admit that this could potentially be overcome by a titratable inducible expression system, but in the context of the other experiments we have performed to meet the journal's resubmission deadline, we have not been able to allocate the significant time that would likely be required for the optimization of such a system.

We have thus utilized the second experimental approach to address the reviewer's concern and to support our model. We generated RNA binding mutants of CELF1 based on previous structural work suggesting that Gly21 and Cys61 in RNA recognition motif (RRM) 1, Gly113 and Cys150 in RRM2, and Gly441 in RRM3 of CELF1 are required for the RNA-binding capacity of CELF1. We generated both individual RRM mutants (Δ D1, Δ D2, Δ D3) where the aforementioned amino acid residues were mutated to alanine and a mutant where all three RRMs were mutated (Δ D1-3). MCF10A cells transfected with the wild-type or individual RRM mutant CELF1 constructs, but not Renilla *Luciferase* transfectants, induced EMT independent of TGF- β treatment, as assessed by relative expression of molecular markers, migration, and invasion (**Supplementary Fig. 3d**). However, ectopic overexpression of CELF1 construct harboring mutations in all three RRMs failed to induce EMT independent of TGF- β treatment (**Supplementary Fig. 3d**). To support the notion that the inability of the CELF1 RNA-binding mutant to induce EMT was due to its lack of interaction with the target drivers, we performed UV crosslinking immunoprecipitation/qRT-PCR experiments. Immunoprecipitation from MCF10A cells transfected with either FLAG-tagged wild-type (WT) or RRM mutant CELF1 (Δ D1-3) with anti-FLAG antibody resulted in effective enrichment of GRE-containing targets only in MCF10A cells transfected with wild-type CELF1. No enrichment of GRE-containing mRNAs was observed from the cells transfected with the RNA-binding mutant CELF1 (**Supplementary Figure 3e**), although this mutant was expressed at similar levels to the wild-type construct. We feel that the aggregate of these experiments effectively address the reviewer's concern.

Issue 5: In Figure 5, the authors demonstrate that overexpression of CELF1 promotes non-invasive breast cancer cell lines to form lung metastasis following tail vein injection in nude mice. The authors should also assess protein levels of GRE-containing targets in the resultant lung metastasis.

Response: Again, this is a fantastic suggestion. To provide exemplars of this, we have evaluated the expression of CELF1, DUSP2, JUNB, SNAI1, and SSBP2 protein in the resultant lung metastases. The data are included in **Supplementary Figure 5b**. Of note, the antibodies used for IHC do not differentiate between human and murine protein, so we do observe some degree of staining within the mouse tissue as well (we did not have the time to repeat these experiments with epitope-tagged human proteins). Nonetheless, each of the GRE-containing targets are overexpressed in the metastatic foci derived from xenografts utilizing CELF1-overexpressing MCF10AT1 cells or β -galactosidase shRNA-expressing MCF10CA1a cells.

Issue 6: One of the most intriguing findings from this study is that CELF1 protein but not mRNA is increased during EMT and in breast cancer specimens. The authors speculate that this occurs at the post-translational level however it would significant increase the overall impact of the manuscript if the authors could provide additional evidence that CELF1 is regulated at the post-translation level during EMT and in cancer specimens by assessing for example CELF1 phosphorylation status. For example they could simply perform a Western Blot with Phospho-specific antibodies, which is commercially available.

Response: This is a very astute observation and would in theory be a fantastic addition to the manuscript. Unfortunately, a phospho-specific CELF1 antibody is not commercially available. Our best option was to query CELF1 phosphorylation status in our primary system. We demonstrate that CELF1 is hyperphosphorylated at both serine and threonine residues, but not tyrosine residues, in mesenchymal MCF10A cells (**Fig. 7d**). That this is consistent with established modes of regulation of CELF1 in other systems (Mol Cell 28, 68-78 (2007)) gives us some insight into the mechanisms at play, but we hope that the reviewer will again agree that a proper and full characterization of this mechanism (P/Ub primacy and order of addition, kinase, E3, DUB) is clearly beyond the scope of what can be presented in a single manuscript.

Minor Points:

1) Upon identification of mRNAs translationally enhanced during EMT, the authors specifically performed analysis on the 3' UTR of these genes to determine whether they harbor a common cis-regulatory element. However, the authors should comment on why they focus on the 3'UTR and also expand on a possible contribution of 5' UTRs towards mRNA translation.

Response: This is clearly an important part of the biology at hand, and we regret that we previously glossed over this concept. Indeed, acknowledging this facet of translational regulation provides significant value in explaining some of the data we present. We have thus included the rationale for focusing on the 3'UTR and also expanded on possible contribution of 5' UTRs towards mRNA translation in the discussion section of the revised manuscript.

2) The authors employ wild-type and mutant ZEB1 3' UTRs as controls in experiments presented in Figure 2C. However, they should provide a brief explanation for the reader as to why wild-type and mutant ZEB1 3' UTRs serve as appropriate controls in this reporter system.

Response: Thank you for your suggestion, this has been included.

3) For consistency, the authors should refer to constructs harboring a deletion of the GRE as either GRE mutant or ΔGRE throughout the manuscript.

Response: Thank you, this has been corrected.

4) From the unbiased polysome profiling analysis performed in Figure 1 and 2 Chaudhury et al. also identify a polysome depletion class of mRNAs upon EMT. The authors should comment upon this and propose potential mechanisms of translational regulation that may exist during EMT in the discussion section.

Response: Thank you for your suggestion, this has been included.

5) On page 6, "UTRS" should be changed to "UTRs".

Response: Thank you, this has been corrected.

Reviewer #2 (Remarks to the Author): Expert in EMT and breast cancer

The manuscript is well presented and the study provided convincing evidence for the translational regulation of EMT using an unbiased approach and highlights the need to consider more than transcriptional approaches to defining EMT. It broadens the recognized role of CELF1, and defines a genetic ordering including a feed forward component that remains to be further characterized. The regulation of CELF1 also remains to be investigated, and this is important in finding ways to gauge the extent to which the CELF1-associated regulon is active in a given scenario.

The methodology appears sound, the study is innovative and novel, and the analytical approaches appear robust.

We are genuinely grateful to the reviewer for his or her positive critique of this manuscript.

Issue 1: In cross-comparing the MCF10A and MCF7 systems, the authors "utilized the rationale that any common regulatory events observed in both models would reflect a regulatory response to TGF-β signaling rather than an event associated with EMT per se." However, it is not known where the EMT blockade exists? It could be downstream of polyribosomal loading / protein synthesis?

Response: This is certainly true, and we did not initially use the best phrasing. While there are caveats associated with any filtering or prioritization approach, an investigator is of course somewhat limited, at least initially, to the level of gene regulation that they choose to dissect. We have rephrased this portion of the manuscript as follows, and hope that the reviewer finds the underlying logic better rationalized:

"Our rationale for comparing the two cell lines was that any polysomal enrichment or depletion event that we observed in either model would directly or indirectly be a result of

signaling emanating from the TGF- β receptor *per se*. However, any such event that was observed in both cell lines could not, by definition, be considered as a candidate that might be required or sufficient to drive EMT in both of the models.”

Issue 2: Snail was found to be present amongst the 72 putative positive regulators. It seems likely that other recognizable EMT drivers should have been present?

Response: This is a very astute observation, and we were ourselves surprised that SNAI1 was the only core driver present following our filtering. Indeed, in the total mRNA data we observed transcriptional upregulation (all values \log_2) of SNAI1 (3.2), SNAI2 (2.6), ZEB1 (1.3), and ZEB2 (0.9). However, of these, only SNAI1 was polysomally enriched (2.8). Polysomal enrichment/depletion of SNAI2 was (-0.8), ZEB1 was (-0.2), and ZEB2 was (-0.7), all values well below our threshold. We note that values for differential expression or polysomal enrichment/depletion for TWIST1, TWIST2, and FOXC2 in our model system were even less remarkable, and Goosecoid wasn't even detected. So had we stuck to traditional profiling, more of the usual suspects would have been present in the analysis. A short acknowledgement of these data has been inserted into the discussion for context and clarity.

Issue 3: The expression of classical markers of breast cancer subtypes does not seem to correspond to expected enrichment, with the exception of PgR and HER2? For example, ESR1 is enriched in basal subgroup, which should be classically ER-negative. Is this because it is expressed compared to normal rather than empirically? It may warrant a comment? Nonetheless the point about CELF1 and the translational targets is made.

Response: The reviewer is absolutely right. The expression of classical markers of breast cancer subtypes in our analysis does not obviously correspond to what one might expect to see enriched given several published studies. The explanation for this is that in our own analysis we were directly comparing relative expression of these mRNAs in primary breast tumors and matched “normal tissue,” which we feel is a better way to highlight dysregulation associated with disease. The rationale for and nature of this comparison has been clarified in the results section of the revised manuscript.

Issue 4: Other modes of post-transcriptional regulation are mentioned briefly in the introduction but not further discussed; for example the regulatory role of non-coding RNAs including microRNA. Upstream regulatory events may have important roles in this pathway, for instance what underlying mechanisms are involved in CELF1 to bind to GRE elements of the RNA regulome (11 genes)?

Response: These are all very salient concerns/questions. For example, when we set the parameters of our analysis software to examine the 3' UTRs of polysomally enriched transcripts, we did this in a fashion that the software should have explicitly been able to find miRNA binding sites. Our rationale for this was simple – “when” we found miR-200 sites, we would have a nice positive control to point to, implicitly validating our approach as we worked down the list of other common motifs. But to our surprise, this didn't happen in the context of our unbiased discovery workflow.

It is our own opinion that the role of members of the *miR-200* family in EMT programs is at this point beyond reasonable contention, and there is little doubt myriad other post-transcriptional mechanisms are involved in this process. The PI has his own very strong opinions about the “micromanager” model of microRNA action, but this is not the appropriate forum for that particular topic to be addressed. Perhaps following the breadth of topic of our introduction, the reviewer is puzzled that we did not address this breadth in the discussion. In light of this possibility, we have included a brief recap in the discussion on this point. This being said, we hope the reviewer will agree that the interplay of these various mechanisms is well beyond the scope of the current manuscript.

The final part of the reviewer’s issue is the possibility that upstream regulatory events may have important roles in this pathway. This is true, but we feel that a full and proper exploration of these events is beyond the scope of this particular manuscript (if this is done properly, one might envision that it would take one or two additional publications, or even one or two full grants to work out. Similar concerns were raised by Reviewer 1, so for the sake of due diligence, we present two new pieces of data that maintain the linearity of the work we present while building the foundation for next steps.

The first piece of data is in regards to the control conferred by the 3' UTR of the GRE-containing transcripts, which might impact mRNA stability, translational initiation, or translational elongation. To differentiate among these possibilities, we assessed the relative expression of reporters fused a subset of our wild-type and mutant 3' UTRs, both in the context of a “standard” reporter (**Fig. 2f**) or under the control of the EMCV internal ribosome entry site (**Fig. 2g**). While transfection of the GRE-containing “standard” 3' UTR reporters into untreated and TGF- β -treated MCF10A cells revealed a significant increase in reporter activity in the mesenchymal state (**Fig. 2f**), this increase was not observed when these 3' UTRs were fused behind a coding sequence under the control of the EMCV IRES (**Fig. 2g**). These results indicate that the control conferred by the GRE-containing 3' UTRs is cap-dependent, and thus likely to be independent of both mRNA stability (verified by qRT-PCR) and translational elongation. Given that phosphorylation of eIF4E has been shown to be required for EMT and metastasis via translational control of a subset of EMT inducers, inclusive of *SNAI1* and *MMP-3* (Robichaud et al. *Oncogene* 34:2032-2042 (2015); Smith et al. *Scientific Reports* 5:18233 (2015)), we can potentially hypothesize, pending further experimental validation, that enhanced CELF1 binding results in enhanced interaction with phosphorylated eIF4E and thus de-repressing the translational block of the GRE-containing mRNAs. We have acknowledged this in the discussion section of the revised manuscript, and expect to further elucidate these mechanisms in future studies.

We also provide evidence that increases in CELF1 protein expression upon EMT are associated with both increased resistance of this protein to proteasome-mediated degradation and increased S/T phosphorylation (**Fig. 7b,c, d**). This mirrors the regulation of CELF1 observed in other model systems (Mol Cell 28, 68-78 (2007)), but again, we hope that the reviewer agrees that properly working through the entire mechanism underlying these phenomena is well beyond the scope of this manuscript.

Issue 5: The number of mRNAs (209) characterized by altered polyribosomal occupancy in this study is not sufficient to claim the study as global.

Response: We respectfully disagree with the reviewer on this specific point. Our ultimate identification of this set of 209 mRNAs was based on an unbiased analysis of both (a) the entire transcriptome and (b) the entire complement of mRNAs enriched or depleted within pooled polysomal fractions. By definition then, we would assert that we performed a global

analysis of polyribosomal enrichment and depletion in the study. The low number of gene products that we ultimately nominated as putative drivers or inhibitors of EMT are a result of *extremely* stringent filtering (q value of 0.03 for the individual relative enrichment or depletion and two standard deviations away from the median in the comparison). When this filtering is compared to that routinely described in transcriptional profiling studies (e.g. a 1.5-fold difference might be considered an acceptable cutoff to assign differential expression), we think the low-ish number of mRNAs that we identify can be better considered in its proper context.

Issue 6. The authors have examined TCGA breast cancer data but not ICGC data. Analysis of ICGC data, which is a much bigger dataset than TCGA, could possibly strengthen findings described in Figure 6, and provide new insights.

Response: It is true that the ICGC collects cancer data from all over the world and therefore has a larger sample size. However, when restricting the search to gene expression (primary site – breast; available data type – EXP-A (expression based on arrays) and EXP-S (expression based on sequencing), which was the goal in our case, the only available data is the Breast Cancer – TCGA, US (please see the figure below). Of the 1,099 donors here, there are 529 EXP-A and 1,041 EXP-S cases, but only 111 matched tumor-normal pairs. We felt that comparison of tumor to normal tissue was the most appropriate comparison for our analysis, and thus limited the data that we present within the manuscript to the comparison of these two sets.

Code	Name	Site	Country	Donors		Available Data Types										
				DCC	All	SM	ONM	SGM	SGY	METHA	METHS	EXPA	EXPS	PDP	NRNAS	ICH
BRCA-US	Breast Cancer - TCGA, US	Breast	US	1,045	1,099	955	1,045	--	--	1,013	--	529	1,041	298	1,026	--
Total				1,045	1,099	955	1,045	0	0	1,013	0	529	1,041	298	1,026	0

Issue 7: From the Methods section (Page 17), it appears that the authors have not submitted NGS data to a public domain; It is important to submit not only total RNA NGS but ribosomal profiling data as well.

Response: The raw sequence files have been submitted to Gene Expression Omnibus with accession codes pending.

Minor points:

1. Should the phrase "invasion metastasis cascade" in line 2 of the paper be "invasion and metastasis cascade"?

Response: Yes, it should be and has been corrected. Thank you for noticing this.

2. In the following sentence "Within MCF10A cells, the observed increase in relative CELF1 protein expression associated with the mesenchymal state (Figs. 2a, 2b) was independent of both changes in relative expression of total CELF1 mRNA and the ribosomal occupancy of these transcripts (Fig. 6b). This does not seem to correspond to Fig 2a, 2b?"

Response: Again, thank you for noticing this. It has been corrected.

Reviewer #3 (Remarks to the Author): Expert in breast cancer progression

This is a dense paper describing translational aspects of the EMT. The authors primarily use MCF-10 and MCF-7 breast cancer lines, with EMT stimulated by TGF- β . This reviewer is not expert in translational regulation and will confine comments to biological relevance. There are probably thousands of papers at this point on the EMT, yet we are still discussing its overall relevance to cancer progression. For a paper to have a significant impact on the field, it must be comprehensive.

Response: We thank the reviewer for his or her analysis of our work in the broader field of breast cancer progression. The opinion of the reviewer here is well taken. In response, we would respectfully assert that in a field this established it is essentially impossible for any individual publication to be truly comprehensive by directly addressing each salient finding of dozens (let alone thousands) of previous studies. Similarly, as a field expands, the likelihood that any individual publication will have a truly significant and comprehensive impact on the field is commensurately decreased. The discussion that the reviewer alludes to is by definition one that remains open, and we feel that our work contributes to this discussion by laying the foundation for future, systematic studies in regards to the finer points of how the program we have defined contributes to cellular plasticity/stemness, chemoresistance, and cancer progression overall. Addressing all of these topics in an already "dense" individual publication is simply not possible.

We are not aware of any previously described studies describing a similar global, and forward approach utilizing polysomal enrichment/depletion analysis to identify distinct mRNAs differentially utilized by the translational machinery in the epithelial and mesenchymal states in an unbiased manner. We respectfully assert that regulation of gene expression at the level of mRNA is comparatively understudied in EMT (and other classically defined hallmarks of cancer), even though the importance of this level of regulation is

beginning to be fully realized in other contexts. As we assert in the manuscript, this regulatory pathway is *completely invisible* in the public datasets derived from major international efforts and consortia that currently serve as the gold standard for genomic reference. This is a sobering finding with potentially very broad implications.

In very few instances was EMT induced by anything but TGF- β or tissue culture. One EGF experiment was shown. These data question the generality of the findings. I would have asked if the same set of genes were found upregulated when EMT was induced in multiple cell lines, under multiple conditions.

Response: The reviewer's assertion that the generality of these findings might have been expanded is well-taken. In this work, we demonstrate that the generality of CELF1 function is conserved among three distinct cellular models of TGF- β induced EMT (MCF10A, MCF10AT1, and HMLE). We next demonstrate that the generality of CELF1 function is conserved between two distinct stimuli (TGF- β in the above-described models and EGF in the MDA-MB-468 model). We next demonstrate that pre-existing CELF1 expression status correlates with molecular markers and functional measures of EMT in these same cell lines (plus MDA-MB-231 and MCF10CA1a models), as does manipulation of this expression. We respectfully assert that for the purposes of a single manuscript not focusing solely on generalization, additional demonstrations of this generalization provide diminishing returns.

Again touching on the theme of comprehensiveness, we are working with a lot of moving parts in this study. Our focus on the generality of CELF1 as the core regulator of this program was a calculated compromise, as conducting similarly thorough experiments with each of the downstream EMT effectors that we identify would provide a bewildering amount of data that would be difficult to fit into an individual manuscript. Would every single component of the pathway that we describe be similarly critical for EMT programs in five to ten distinct experimental systems? It is unlikely, and we are patiently unwilling to assert that this is the case here in our response to the reviewer or within the context of the manuscript.

In support of these assertions and to move towards addressing the reviewer's point, we now provide a limited analysis of some of the effector genes that we identified in the initial MCF10A model system in the context of MDA-MB-468 cells treated with EGF (**Supplementary Figure 4a**). These data indicate increased expression of EGR3, JUNB, SNAI1, and SSBP2 protein upon treatment of these cells with EGF. Notably, FOSB protein expression is unchanged in this context. This is perhaps not unexpected, given that our epistasis analysis (**Figures 4e, f, and 8**) places the FOSB transcription factor genetically upstream of CELF1 in a forward amplification loop. Activation of distinct signaling pathways (SMAD vs JAK/STAT in this context) can be expected to result in distinct responses, both transcriptionally and even further downstream, that even so may ultimately lead to a similar phenotypic outcome. Thus, although many of the "details" of what occurs may be specific to a particular condition or stimulus, we feel that we have provided strong evidence that the role of CELF1 (and ONLY the role of CELF1) in EMT is conserved and generalizable in multiple systems in response to multiple stimuli.

The other question is the relevance of EMT. The authors show that E-cadherin and other markers change in expression, but there is only one functional metastasis experiment using one of the genes. Motility and invasion data could be easily added. Other investigators view EMT as a marker of stemness, of cellular

transdifferentiation, and of plasticity, and it would have been of great interest to know what happens in these arenas.

Response: We are somewhat confused by this comment as motility and invasion data are included as a component of each of our measures of EMT (with the exception of the introductory Figures 1a and 1b). Please note the bar graphs and micrographs present under each of the immunoblots in **Figures 3, 4, and 5**.

We are similarly enthusiastic to the reviewer in regards to the potential relevance of our findings to stemness, cellular transdifferentiation, and/or plasticity. Yet, we respectfully submit that the proposed experiments are beyond the scope of the current work. We plan to address these issues in the near future, but we hope that the reviewer will concur that if done properly, dissection of each of these aspects has the potential to comprise one or more stand-alone manuscripts.

The human tumor data has so few Gr III tumors as to be non-informative.

Response: The resource that we used for this dataset was the best available to us and provided multiple benefits in regards to analysis bias. A broad range of tumor types and grades was present, in random orientation, on the same glass slide. The benefits of this resource were then that variation in IHC staining was thus minimized, and the pathologist scoring the staining was given a broad and diverse range of tumors to score (which of course means a more uniform and normalized range of relative scoring). We assert that this is the best insurance against bias in this type of analysis.

Ultimately, although we agree with the reviewer that more would have been better, we respectfully disagree that the numbers we include are low enough that they should be dismissed out of hand. The data derived from these Grade III tumors indeed fits our model in a fashion that is supported by rigorous statistical analysis. Were either the numbers of these tumors or datapoints that they provided insufficient to support the argument that we make here, the statistical test would have returned an insignificant value.

Finally, please do not use the term "master regulator".

Response: This was perhaps an unconscious by-product of heavy and repeated exposure to a certain faculty member during the PI's graduate schooling. I distinctly remember a certain Nobel Laureate piping up during one of this faculty member's seminars saying ("We have those too – they're called transcription factors." The reviewer is correct – the term is terrible and has been replaced.

Reviewer #1 (Remarks to the Author)

The authors have satisfactorily addressed my comments and this very nice paper is ready for publication.

Reviewer #2 (Remarks to the Author)

The revised manuscript contains significant additional data.
My Issues are largely addressed, with the exception of the following:

Issue 1: In cross-comparing the MCF10A and MCF7 systems, the authors "utilized the rationale that any common regulatory events observed in both models would reflect a regulatory response to TGF- β signaling rather than an event associated with EMT per se." However, it is not known where the EMT blockade exists? It could be downstream of polyribosomal loading / protein synthesis?

Response: This is certainly true, and we did not initially use the best phrasing. While there are caveats associated with any filtering or prioritization approach, an investigator is of course somewhat limited, at least initially, to the level of gene regulation that they choose to dissect. We have rephrased this portion of the manuscript as follows, and hope that the reviewer finds the underlying logic better rationalized:

"Our rationale for comparing the two cell lines was that any polysomal enrichment or depletion event that we observed in either model would directly or indirectly be a result of signaling emanating from the TGF- β receptor per se. However, any such event that was observed in both cell lines could not, by definition, be considered as a candidate that might be required or sufficient to drive EMT in both of the models."

Reviewer: I still take issue with the wording here, because a coordinated post-translational block on EMT-induced proteins could still mean that factors that were common to both models may indeed be "a candidate that might be required or sufficient to drive EMT in both of the models", in the appropriate context. Obviously the approach has been successful, and this is simply semantic, however I would like the authors to choose a phraseology that accommodated this hypothetical issue. Perhaps a terminology that rather indicated that any polysomal enrichment or depletion event could be associated with the differential EMT response. i.e. focus on the implications of things being different rather than the assumptions that I believe shouldn't be made if they are not different.

Issue 6. The authors have examined TCGA breast cancer data but not ICGC data. Analysis of ICGC data, which is a much bigger dataset than TCGA, could possibly strengthen findings described in Figure 6, and provide new insights.

Response: It is true that the ICGC collects cancer data from all over the world and therefore has a larger sample size. However, when restricting the search to gene expression (primary site - breast; available data type - EXP-A (expression based on arrays) and EXP-S (expression based on sequencing), which was the goal in our case, the only available data is the Breast Cancer - TCGA, US (please see the figure below). Of the 1,099 donors here, there are 529 EXP-A and 1,041 EXP-S cases, but only 111 matched tumor-normal pairs. We felt that comparison of tumor to normal tissue was the most appropriate comparison for our analysis, and thus limited the data that we present within the manuscript to the comparison of these two sets.

Reviewer: Indeed, surprisingly, the ICGC only has 3 breast datasets and only one with matched tumour and normal. This is surprising given the larger number of datasets seen in other cancer types. Are the authors restricted to ICGC/TCGA though, are there not other datasets that allow this comparison to be made? Oncomine lists many breast cancer datasets with matched tumour and normal.

Reviewer #3 (Remarks to the Author)

The authors have attempted to revise this manuscript to please reviewers of highly divergent interests. The paper is clearly most relevant to researchers interested in post-transcriptional regulation. As for general breast cancer biology, the paper reads more solidly.

1. Fig 5A simply shows that TGF- β /EGF induces EMT in epithelial lines and that the gene of interest is co-regulated. The best data in the Figure is shown in Fig 5B, where knockdown or overexpression of CELF1 influences motility and invasion. Please revise how this figure is laid out, so that the bar graph has an x-axis label stating what conditions are being tested. The pictures below the bar graph are too small to see and can be moved to Supplemental Data.
2. A metastasis model in two sublines of the MCF-10A system are shown. Most metastasis papers in medium/high impact journals require two models so this checks the box, although it would have been better to include completely independent model systems.

Reviewer #1 (Remarks to the Author)

The authors have satisfactorily addressed my comments and this very nice paper is ready for publication.

Response: Thank you.

Reviewer #2 (Remarks to the Author)

The revised manuscript contains significant additional data. My Issues are largely addressed, with the exception of the following:

Response: Thank you.

Current Issue 1: In cross-comparing the MCF10A and MCF7 systems, the authors "utilized the rationale that any common regulatory events observed in both models would reflect a regulatory response to TGF- β signaling rather than an event associated with EMT per se." However, it is not known where the EMT blockade exists? It could be downstream of polyribosomal loading / protein synthesis?

Initial Response: This is certainly true, and we did not initially use the best phrasing. While there are caveats associated with any filtering or prioritization approach, an investigator is of course somewhat limited, at least initially, to the level of gene regulation that they choose to dissect. We have rephrased this portion of the manuscript as follows, and hope that the reviewer finds the underlying logic better rationalized: "Our rationale for comparing the two cell lines was that any polysomal enrichment or depletion event that we observed in either model would directly or indirectly be a result of signaling emanating from the TGF- β receptor per se. However, any such event that was observed in both cell lines could not, by definition, be considered as a candidate that might be required or sufficient to drive EMT in both of the models."

I still take issue with the wording here, because a coordinated post-translational block on EMT-induced proteins could still mean that factors that were common to both models may indeed be "a candidate that might be required or sufficient to drive EMT in both of the models", in the appropriate context. Obviously the approach has been successful, and this is simply semantic, however I would like the authors to choose a phraseology that accommodated this hypothetical issue. Perhaps a terminology that rather indicated that any polysomal enrichment or depletion event could be associated with the differential EMT response. i.e. focus on the implications of things being different rather than the assumptions that I believe shouldn't be made if they are not different.

Response: We will initially like to thank the reviewer for acknowledging that we have addressed most of the concerns raised during the initial peer review. As per this advice, and

because of editorial length requirements, we have now reduced this rationalization to the following: "We rationalized that any event commonly observed in both cell lines could not be associated with the differential EMT response in these models."

Current Issue 2. Original Issue 6. The authors have examined TCGA breast cancer data but not ICGC data. Analysis of ICGC data, which is a much bigger dataset than TCGA, could possibly strengthen findings described in Figure 6, and provide new insights.

Initial Response: It is true that the ICGC collects cancer data from all over the world and therefore has a larger sample size. However, when restricting the search to gene expression (primary site - breast; available data type - EXP-A (expression based on arrays) and EXP-S (expression based on sequencing), which was the goal in our case, the only available data is the Breast Cancer - TCGA, US (please see the figure below). Of the 1,099 donors here, there are 529 EXP-A and 1,041 EXP-S cases, but only 111 matched tumor-normal pairs. We felt that comparison of tumor to normal tissue was the most appropriate comparison for our analysis, and thus limited the data that we present within the manuscript to the comparison of these two sets.

Indeed, surprisingly, the ICGC only has 3 breast datasets and only one with matched tumour and normal. This is surprising given the larger number of datasets seen in other cancer types. Are the authors restricted to ICGC/TCGA though, are there not other datasets that allow this comparison to be made? Oncomine lists many breast cancer datasets with matched tumour and normal.

Response: We concur with the reviewer in regards to the potential of analyzing the Oncomine or other data sets with more matched breast cancer data sets. However, the initial recommendation was to analyze ICGC data which did not reveal any additional information beyond the TCGA analysis. We hope that the reviewer will concur that if done properly, the analysis will take a prolonged time and delay the publication of the current manuscript, particularly because our bioinformatics collaborator has been somewhere in the Amazon for the last month. We have provided evidence on multiple levels that the changes we observe are at the level of translation, and whereas this additional analysis will further validate those findings, we respectfully submit that it will not provide any additional new information.

Reviewer #3 (Remarks to the Author)

The authors have attempted to revise this manuscript to please reviewers of highly divergent interests. The paper is clearly most relevant to researchers interested in post-transcriptional regulation. As for general breast cancer biology, the paper reads more solidly.

Response: Thank you.

Issue 1. Fig 5A simply shows that TGF- β /EGF induces EMT in epithelial lines and that the gene of interest is co-regulated. The best data in the Figure is shown in Fig 5B, where knockdown or overexpression of CELF1 influences motility and invasion. Please revise how this figure is laid out, so that the bar graph has an x-axis label stating what conditions are being tested. The pictures below the bar graph are too small to see and can be moved to Supplemental Data.

Response: We appreciate the reviewer agreeing that Figure 5B presents relevant data. We however, respectfully disagree with the reviewer's suggestion of reformatting the figure. In every experiment where we have assayed for EMT it has been by (a) immunoblotting for the epithelial cell marker, E-cadherin, and mesenchymal cell markers, N-cadherin, vimentin, and fibronectin; and (b) by scoring *in vitro* migration and invasion both using Cultrex 96 Well Cell Migration and Invasion Assay kits and by imaging more traditional transwell inserts coated with basement membrane matrix. All the resultant figures have been laid out to show immunoblot results at the top, the quantification of *in vitro* migration and invasion in the middle, and the images of *in vitro* migration and invasion at the bottom. Hence, all these figures use the same x-axis which is labeled at the top of the figure in each case. Such a pattern helps us in having a uniform formatting across figures for easier understanding and interpretation of the reader, and also in showcasing the result from the entire experiment. In addition, our supplementary figures are already very busy and including any additional data there is virtually impossible. So the bar graph in Figure 5B does have an x-axis at the top of the figure and moving the same level that to the bottom will be redundant. We hope that the reviewer will concur.

Issue 2. A metastasis model in two sublines of the MCF-10A system are shown. Most metastasis papers in medium/high impact journals require two models so this checks the box, although it would have been better to include completely independent model systems.

Response: Thank you. We agree with the reviewer that validating our model in additional *in vivo* models are required, something that will definitely be one of our focus moving ahead.